# Common Space and Behavior at the Border between Slum and Metropolitan Area: The Case of "Catambor" and "Alvalade"

**Yannick Oliveira** [1,*] , **Suguru Mori** [2] **and Rie Nomura** [2]

1 Graduate School of Engineering, Hokkaido University, Sapporo 060-8628, Japan
2 Faculty of Engineering, Hokkaido University, Sapporo 060-8628, Japan
* Correspondence: yannick.oliveira.092@gmail.com or
yannickvascodomingosde.oliveira.p9@elms.hokudai.ac.jp; Tel.: +81-80-9007-4192

**Abstract:** Composed of modern city centers and numerous slum patches in deep proximity and coexistence with one another, the Sub-Saharan African urban landscape is often perceived as chaotic or unorganized in nature due to the inconsistencies in the urban layout derived from the spontaneous occupation in said slums, in Angola, known as "Musseque". This article focuses on the border between the "Musseque" of Catambor and the Alvalade Neighborhood as a point of interaction of both realities, influencing users to either adjust their activities to the streets or adjust the streets to their needs. With the purpose of understanding the streets' environmental behavior settings and purpose improvements while preserving said environmental behavior, this study uses the behavior mapping method to identify the users' stationary activities and then groups them by zone and occurrence, followed ultimately by a series of interviews. The results uncovered a degree of self-intervention by the users, ranging from the establishment of "commercial spots" on the street to "the setting of places to sit and gather" to accommodate for the lack of services, defying the intended purposes of the streets and prompting a deep understanding of what guides the decision-making processes and what the users consider valuable to the built space.

**Keywords:** musseque; slums; behavior settings; urban intervention; environmental behavior; street improvement

## 1. Introduction

### 1.1. Background

Public Space and Streets

Public space as a notion has been defined by Jurgen Habermas as "a physical space where the public domain is expressed in space" which suggests a node to an apparent pseudo-ownership of the public space by its users and the role of community in the development of its perceived character and identity, as expressed in its physical elements as well as the non-physical ones, such as history and culture [1]. Streets, more specifically, can be defined as the stage where all this interaction takes place, from the access to buildings, to movement, or locomotion between different points either within a city or in and out of a city [2]. That influence on users' daily lives places streets as an important component for the definition of one's perception about a city being the place where human exchanges and relationships are manifested alongside the diversity of use and vocation of each place, as well as the contradictions of society itself [3]. That idea, coupled with the current urban expansion trend and slum growth [4], supports the need for rethinking our idea of street planning toward a more inclusive, resilient, and sustainable approach [5], as suggested by the United Nations Sustainable Development Goals (SDGs).

Mathew Carmona made the case that both the role of culture and the specific needs of the people of a specific area bared a higher weight in planning and designing public space than the uniform approach of globalization [1]. Making the case that although

common structures and characteristics can be identified in streets regardless of place, the notion of common space and its limits and characteristics can be interpreted differently by different cultures, suggesting that cultural changes and populational shifts can bear a different impact on the transformation of street environments within a city. The physical environment in public spaces can be seen as properly or improperly adjusted to the population it responds to, depending on how well they serve the purpose for which they were planned, whichever that might be. The case is made that the exterior space can be considered positive when providing specific functionality with a clear definition of boundaries, whereas the space is considered negative when it does not encompass a clear definition of boundaries and specific functionality [6,7]. This suggests, in turn, that the adaptability of the environment is linked to how well it adjusts to the user's needs without breaking apart from its intended functions. Because of the nature of the relationship between the people and the inhabited place, manifested in every interaction within the physical environment, understanding behavior as an integral part of the function of the place with temporal and spatial boundaries should provide a suitable basis for the analysis and evaluation of how well the space adjusts to the users, as suggested by Barker [8–10].

The study found that from the users' perspectives, public space is not out of range regarding semi-permanent or even permanent self-made design changes to cope with the lack of services. It is possible to say, at least to a degree, that the users themselves would view the public space near them as more semi-public in nature, making them unafraid and unrestrained to make changes to the street itself. The scale to which one perceives their own residential area, be it the neighborhood, closest streets, or simply the closest buildings can have a clear impact on one's own residential satisfaction [11].

### 1.2. Economic Development and the Growth of Slums

Looking at the relationship between general urban growth, economic development, and the growth of the slum population in the world, there is a noticeable connection between those elements and the emergence and growth of slums in the world in recent decades [4].

The decade from 2000 to 2010 registered a high influx of people into urban areas, having the urban population cross the 50% mark by 2008. Such growth is projected to raise the urban population up to 4.9 billion by 2030 while shrinking the rural population by 28 million. This growth is being proven hard to deal with for urban planners, as it creates an imbalance between housing development and the housing demand to be met, which by itself increases housing costs to speculative prices, rapidly absorbing the poorer population into slums and being the only affordable housing solution for some [12].

The rapid growth of cities has been linked to the birth of slums on many different occasions, with the most prominent cases being found in Sub-Saharan Africa and Southern Asia [13]. The connection between rapid urban growth and slum development is particularly apparent when we refer to Sub-Saharan Africa [14]. With only 43% of its population living in cities, as a continent, Africa is the least urbanized region of the world. That, coupled with the fact that Africa and Asia are currently projected to accommodate close to 90% of the increase of the world's urban population by 2050 [15], suggests that the potential growth of slums, in general, is much higher in those areas.

Despite the rapid growth rate in the urban population, the proportion of slum dwellers relative to the urban population, in general, is, in fact, in decline, but that decline does not translate into a reduction of slums in general or even a reduction of the slum population. From 1990 to 2010, the population living in slums in Sub-Saharan Africa went from 70.0% to 61.7%, signaling a reduction relative to its total urban population, while the number of slum dwellers went from 102.6 million to 198.1 million [16]. This suggests a high population growth within slums but an even higher urban population growth in general, inferring that despite the relative decline in the slum population growth rate, that does not translate to a resolution of the slum problem; it instead reaffirms the disproportional populational growth within cities overall, nonetheless prompting the increase in the slum population.

The Angolan "musseque" is a manifestation of that phenomenon. Usually associated with poverty and misery, it draws parallels to shantytowns, in general, and is commonly linked to the Brazilian "favela". This stems from the need and search for proper housing, access to water, and electricity; quicker access to hospitals, schools, and commercial stores; and other living amenities commonly and properly associated with city life in the modern day [17,18]. From that disproportional and increasingly divided and unsustainable urban growth, we have an increasingly pertinent need for a more sustainable look toward our communities, as pointed out by the 11th United Nations Sustainable Development Goal (from here on referred to as SDG11), when addressing housing and infrastructure, and in its efforts toward strengthening the preparedness and resilience of cities in order to ultimately lift slum dwellers out of poverty, promoting inclusion and reducing inequalities in the process [5].

### 1.3. Purpose and Significance of the Research

The investigation arises from a necessity to look at the slum and city environment not as opposites but as two different environments that, while having a proximity relationship, also have a massive gap between them regarding the quality of life and differences in the architectural space and urban landscape that ultimately translate to differences in quality of life and perceived social status across both realities.

In the pursuit of creating a more sustainable street environment, the study attempts to classify and shed a light on the environmental behavior itself, for they teach us what the community finds useful and necessary, but the environment simply cannot provide.

This study hypothesizes that in order to cope with the inadequacies of the streets regarding the daily needs of slum dwellers, that is, to move, mingle, and even for commercial purposes in general, design changes do occur on the streets promoted by the users themselves in order to accommodate for naturally occurring behavior, either on a conscious or subconscious level. This results in a completely unique set of behaviors on the street, changing the environment in the process.

The significance of the study itself lies in the notion that there is a gap to bridge between the cultural and socioeconomic needs of the users and the design and intended functions of the streets upon inception.

Remali points out that residential satisfaction is one of the main points behind the idea of pursuing a life near the city center. That pursuit of residential satisfaction can be considered one of the main motivators of the continuous attempts of improving the living environment conducted by the users themselves. Many of those interventions come in the form of street improvements, giving grounds to the role of the streetscape in shaping the users' qualitative assessments of the social environment, even if through a purely subjective lens [19].

However, the theory of urban ecology by Park and Burges points toward a much deeper analysis and proposes that similarly to natural ecosystems, the division of the urban space could be attributed primarily to the pursuit of urban resources, such as "land", which leads to a competition between groups and, ultimately, to the grouping of people with the same or similar social characteristics due to being subjected to similar social challenges [20].

Similar to the circular economy model and in accordance with the current United Nations SDGs, the implementation of policies promoting social inclusion and the involvement of the community itself can be considered of the utmost importance for solving problems in the urban context as a part of urban regeneration itself [21]. However, the implementation in areas with a completely unplanned design, such as African slums, where the redesigning of complete areas of a city is often considered the primary go-to by urban planners, can pose unique challenges and raise questions about which elements of the physical environment should be preserved instead of demolished, and which environmental features should be given proper attention given that, in general, the overall perception of comfort is at stake.

As a colonial city, Luanda was built to sustain the population and the problems existing by the time of its inception, but over the years, over-population has put pressure on the city,

increasing the size of the city itself while the re-planning of the city, in terms of occupation politics adapted to its now ever-growing size, never happened. That has led to a problem with many urban asymmetries throughout the city, leading to many new zones of irregular settlements throughout the city, called "musseque".

In the following decade after the end of the civil war (1975–2002), the county's GDP grew at rates between 0.9 and 15%, averaging at 9.72% in the span of 10 years (2022–2012). In the same period, the city found itself expanding, with the population going from 3.2 to 6 million, having reached 9 million in 2022 [22]. That state of constant expansion is often credited with the rise of asymmetries already existing throughout the city.

This study attempts to address the inadaptability of the streets at the border between slum and metropolitan areas to account for the uncovered behavior settings, which could be a key factor for the perceived misuse of the streets in the research area, as noticed from the existence of selling spots in areas that would otherwise be destined for circulation, and the persistence of particular types of stationary behavior in the middle of the road as the section of uncovered behavior discusses further.

*1.4. Slums and the Musseque Problem in Luanda*

According to the UNHabitat, a slum household is defined as "an individual or a group of individuals living under the same roof in an urban area, lacking one or more of the following amenities" (Table 1).

**Table 1.** Slum Definition by Missing Amenities.

| Missing Amenities | Definition |
| --- | --- |
| 1—Durable Housing | -A permanent structure providing protection from extreme climatic conditions |
| 2—Sufficient Living Area | -No more than three people sharing a room |
| 3—Access to improved water | -Sufficient, affordable, and can be obtained without extreme effort |
| 4—Access to improved Sanitation Facilities | -Private toilet, or a public one shared with a reasonable number of people |
| 5—Secure tenure | -De facto or de jure secure tenure status and protection against forced eviction |

With the current urban development rates in Sub-Saharan Africa, those settlements seem to be directly associated with the current rates of urban growth [23]. Moreover, given the unplanned nature of the settlements, there is a vast number of different characterizations focused on understanding the types of layouts, formation patterns, and environmental characteristics, depending on the country. However, according to Baross's OBIP (Occupy, Build, Infrastructure, and Planning) logic as a method (Figure 1), characterization based on land occupation seems to perform the best job encompassing the differences between formal urbanization and the informal development directly associated with slums [24].

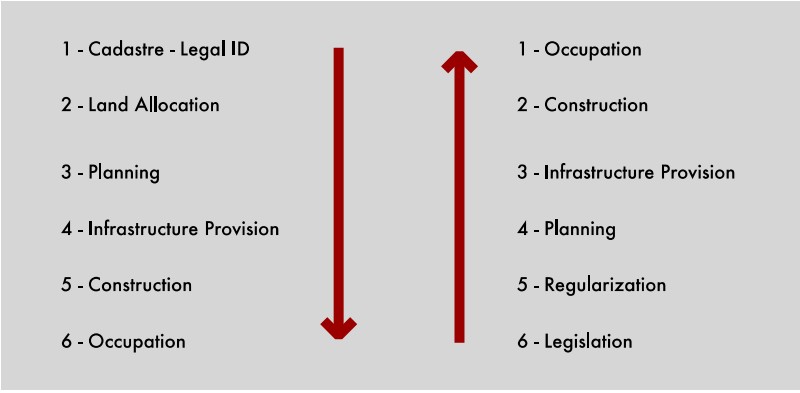

**Figure 1.** Land Occupation Method [24].

Luanda's metropolitan area accounts for about 30% of the country's population estimated at about 9 million inhabitants, making it the largest and densest city in the country,

compared to 0.59 million at 8.26% of the country at the time of independence (1975) [25]. This sudden population growth has been attributed by 42% to natural growth and 58% to inner city migration, returning refugees, and migration from other provinces, with the role of major instigator being attributed to the Angolan civil war (1975–2002) [26], which accounted for a major displacement of people from its inception to end [15,27]. That displacement, fueled by the pursuit of a better life, has led to a massive population increase in the city and, subsequently, the surge of many different "musseque" in the same period [24].

The term "musseque" is the most common denomination of a slum in Luanda, which, coming from the Kimbundu "Mu" (place) and "Seke" (red sand, or simply sand), stands for "place of red sand" or simply "places of sand" in reference to the lack of infrastructure, asphalt on the roads, and as a reference to the natural soil in Luanda which is mostly composed of red sand. The term itself is commonly addressed with a derogatory connotation, with the musseque residents on the receiving end of it [16]. This can be attributed to the historical background of the city at the time of independence, the cultural and political background, and the perceived living reality inside the musseque with poor access to amenities and services [28]. "Development workshop" makes a characterization of the distinct types of musseque according to (1) structure of the settlement, (2) timeframe of appearance, (3) security of the tenure, (4) quality of construction, and (5) access to infrastructure [29], grouping them into the following categories:

- Transition Musseque—Located close to the city center to make use of the services and amenities that come from being close to the city; often smaller in area but denser than the others due to the limited expansion possibilities;
- Ordinated Musseque—Tends to be a more organized urban layout in comparison to the other categories due to the occupation pattern, as it often attempts to follow the urban layout of the neighboring popular neighborhoods;
- Old Musseque—As the name suggests, this is the oldest form of musseque, dating from before independence, and are recognized to have grown simultaneously with the city;
- Peripheric Musseque—These are the opposite of the transition musseque regarding the number of services as they tend to be, as the name suggests, on the periphery of the city, making it much more isolated from the center.

Due to its direct proximity to the center, the abrupt change in the urban layout near the border with Alvalade, and the specific nature of behavior on the streets uncovered from the early research, Catambor was identified as a fitting area for the research.

## 2. Materials and Methods

### 2.1. Research Area

The research was conducted on the "Musseque of Catambor" in the city of Luanda, Angola (Figure 2), with the focus being the border between the musseque itself and one of the neighborhoods that border it, the "Alvalade neighborhood". Located in the Maianga district, both neighborhoods are near the city center. "Catambor" is what is called the "Colonial Musseque", which, as suggested, derives its early settlements formed in the aftermath of the Angolan colonial period and expands itself from that point following the different migration waves.

There have been different attempts to provide a definitive solution to the Catambor neighborhood, with the most recent proposal including the complete destruction of the neighborhood to give place to a new modern neighborhood by the year 2012.

The work attempts to characterize the existing behavior settings, focusing specifically on the human behavior and spatial analysis of the streets and alleys that border both neighborhoods directly, or streets that connect with "Helder Neto Street", "P. Marien da Mata Avenue", and "Dr. Egas Monis Street", as those constitute the border between both neighborhoods.

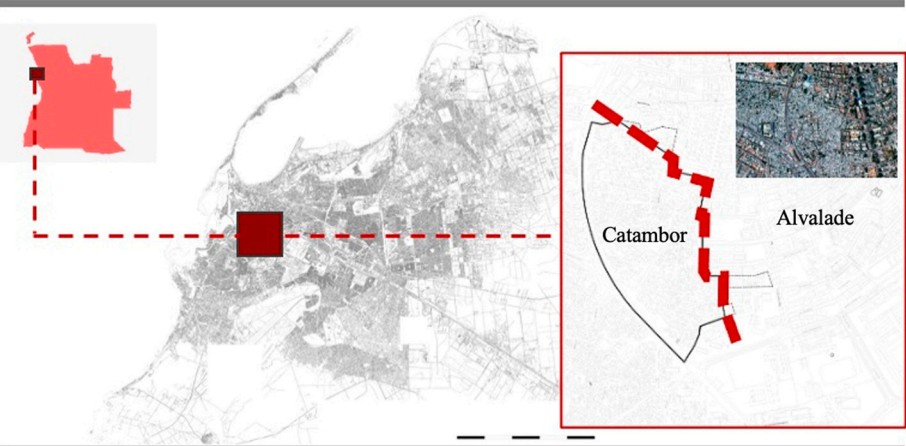

**Figure 2.** Research Area.

Built in the late 1950s and stemming from the Arab "al-batat", meaning "slab", "pavement", or "stone pavement", the Alvalade neighborhood mirrors the similarly named neighborhood in Lisbon. The neighborhood reached the 1970s as one of the most well-regarded and sophisticated neighborhoods in the city. Located in the "Maianga" district at the core of the city and home to many important buildings of the city, such as "The Karl Marx Movie Theatre" and the "Alvalade Olympic Pool", the neighborhood did fall into the category of desirable places to live in, even if only in close proximity to it given the relative standard of living at the core of the city.

As a Transition Musseque [30] at the heart of the city of Luanda, "Catambor" exists in close proximity to a wide array of different services near the city center, potentially mitigating the absence of some of the services and amenities faced by the slum dwellers, a factor that as indicated by the literature, deeply influences the birth of the musseque in the first place, as well as influencing some of the behavior settings that this research would ultimately uncover.

Although the study was focused on the area that constitutes the border between both areas, the analysis does have a larger focus on the catambor musseque, for its environmental characteristic (Table 2), and its influence on the uncovered behavior settings. That focus on the border, is due mainly to two reasons: the first one is the clear difference between those two realities regarding the use of common space, most specifically street usage, and second, for the specific set of behavior settings previously found in the area that differed from the streets inside both neighborhoods. This created a situation where the majority of permanent users or users with direct impact on the interventions of the street were from Catambor, and aside from Catambor residents themselves, many of the users visiting to receive services were from Alvalade.

**Table 2.** Details of the research area by 2012. (Angolan National Institute of Statistics—2012) [31].

| Area Name | Catambor |
|---|---|
| Location | Maianga District |
| Period of Creation | Independence (1975) |
| Population | 3500 |
| Number of Households | 875 |
| Houses with Electricity | 75% |
| Houses with water | 2% |
| Cement Constructions | 30% |
| Wood Palm and others | 60% |
| Materials | Varied |
| Structures and Amenities across the street | Convenient Stores, Retail Shops, Food Kiosks, Drink Shops |
| Construction materials | Asphalt, Concrete, Sand |
| Height of Structures | 1–2 Stories |

These findings posed the question of "what was in the base for those specific behavior settings", "is it an extension of the specific culture of the place", "is it influenced by the characteristics of the physical environment", or potentially a mix or both, as further research would come to confirm.

### 2.2. Data Collection

For an in-depth assessment of the behavior at the border, the survey was divided into two main parts: The first one consisted of the behavior mapping method to properly identify people's activities, which were then grouped into different behavior setting categories and sorted into different zones according to occurrence [8], supported by a photographic survey and followed by a series of interviews in order to obtain a better glimpse into the reasoning for the existing behavior, what guides the decision-making processes, and what the users consider valuable to the built space.

The behavior mapping took place between 20 July and 14 August 2020 over the span of 12 weekdays and four weekends, where the area was divided into four different segments based on natural limits such as the end of a road or the change of its nature, and due to the length of the survey area itself. Those segments were then surveyed separately for the timeframe of three weekdays and one weekend per place, specifically. The observations were conducted to record stationary activities on the base map using the behavior mapping method in a process repeated three times during the day, from 9:00 to 10:00 in the first period, from 12:00 to 13:00 in the second, and the third period from 15:00 to 16:00. This was to provide a better understanding of the dynamics of the behavior of the place across the day, the type of changes that occurred depending on the period, and the motivations for those changes.

The mapping was then followed by a series of interviews conducted over the span of one week, between 27 September and 2 October 2020, and mainly between 12:00 and 16:00 since most behavior settings were concentrated at that time, specifically. The interviews were designed with the intent of obtaining in-depth knowledge of the type of activities of the users, and regarding frequency, motivations, and overall impressions of the physical environment, to serve as a complement and support to the findings of the behavior mapping phase. The design of the interviews, just like the questionnaire on the pre-survey, was composed of four parts: The first one was composed of personal information, such as age, gender, main source of income, and residence status for an accurate characterization of the structure of the sample and a better understanding of the type of population and main trends about the users of the street. The second part was based on motivations for the users to come to the specific spot of inquiry. The third part focused on obtaining more in-depth information about the most common activity named in the previous section. Lastly, the fourth and last part of the interview focused on acquiring data related to their impressions about the specific place of inquiry itself, containing questions such as: what they would find attractive about the place itself from the user's standpoint regarding the physical environment, points of interest, and points of improvement about the area.

The interviews function as a support for the qualitative nature of the behavior analysis itself [32,33]. Here, the objective is to understand based on the behavior settings uncovered in the behavior mapping phase and according to the sample: "To which extent those behavior settings tend to be repeated by varied types of users"; the "main motivations and factors that potentially increase the occurrence of those behavior settings", in the wider area in general, and in the survey spot more specifically; the "variability of those behavior settings from time and place"; "which impressions do the users have about their time at the specific survey spot according to the use cases; and, ultimately, what type of impressions and improvement suggestions they could potentially have for the specific survey spots.

Finally, after the data acquiring process, the different data was processed, coded, and analyzed to reach a conclusion as to how to properly characterize the behavior settings in the research area, "what role does the physical environment play into influencing

behavior", and "which sort of changes do occur to the physical space that result from the found behavior settings".

### 2.3. Characteristics and Places of Focus Regarding Street Usage

In addition to the location of behavior and physical characteristics, several specific points of reference were identified in different spots of the research area that functioned as a base for the analysis of the physical characteristics and as an early basis for the understanding of physical aspects of the environment guiding the existing behavior.

In the area described as P1 (Figure 3), three things in the spot seemed to guide the main behavior settings: the "Roulotte", used as a Kiosk, as it is very common in the entire research area; the Football table; and the Board for board games, essentially making it a game hub for residents and visitors, making it primarily an area for people to have fun or amuse themselves.

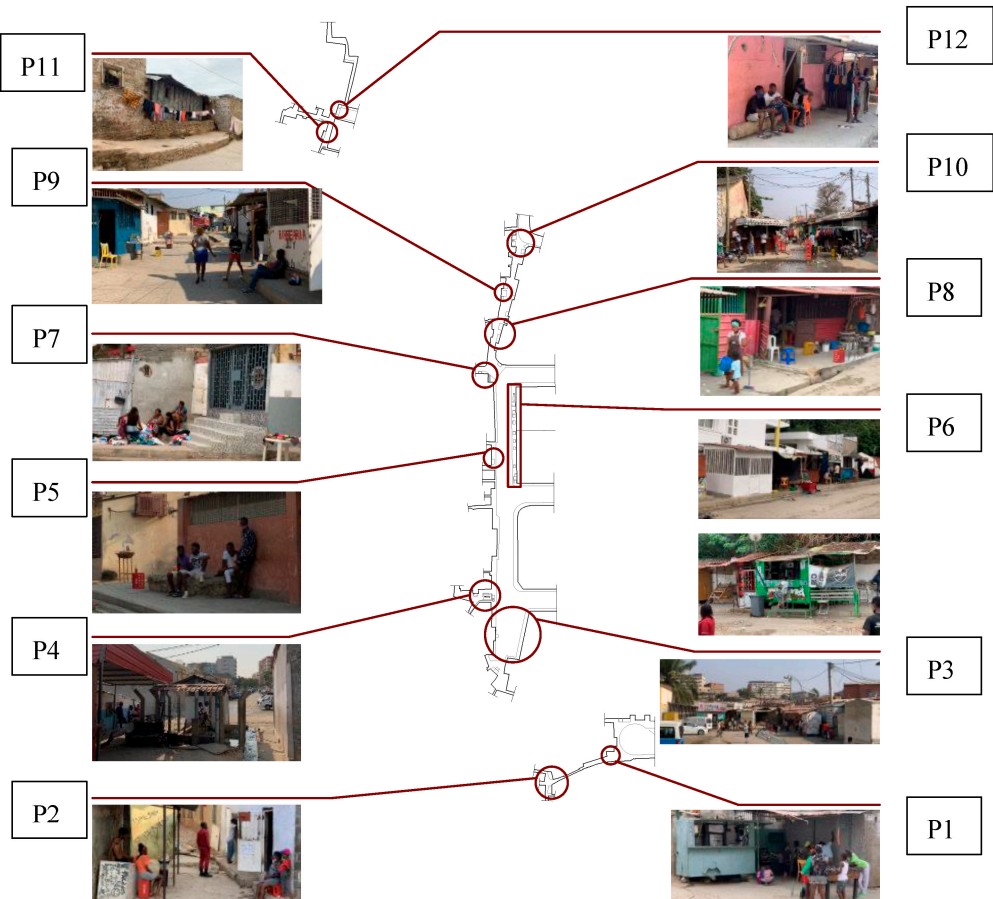

**Figure 3.** Points of Reference.

In the P2 area, the activities can vary from being a place with a small commercial spot to a place for sitting and relaxation, depending on the day of the week and time of the day, with an emphasis on the covered area with the broken fridge laid on the floor, functioning either as a place to sit or a table to put the products as an expository stance when being used as a selling spot.

The zone described as P3 is a very large part of the road that exists likely due to irregularities in the urban design and, simply due to sheer size, it is often utilized as a sports court mainly for football and basketball; for that reason, it attracts many people either to watch and enjoy the game or to buy different goods in the selling spots.

P4 is mainly a place where people tend to gather by the shadows and with space to sit. The place was born from a former service hub called "Chafariz", a small construction element that exists for people to receive water due to the insufficiencies of the distribution

network, and although, as of the time of the survey, it was not operational, the place still retained the functions related to social gathering.

In the zone denominated as P5, we find a big stone on the floor that seemed to function as an important spot to gather for many groups of people, and although it occupies the sidewalk, its geometry and position allow for this recurrent behavior despite the variations on the frequency of use and occupation depending on the time of the day.

In the case of P6, the Sidewalks are completely utilized as a retail zone for the residents, prompting different mitigations to the absence of the sidewalk, such as sitting and waiting for food directly on the road.

In P7, the staircases can be often used to sit down and chat, as the land is on a remarkably high slope in some places, which can justify the specific uncovered behavior.

P8 is defined as an area where the household makes use of the common space in the street as part of their houses for commercial purposes, cooking and selling food for passersby and for those coming from the companies at the center of the city during lunchtime.

In the case of the space defined as P9, the height of the step allows for a person to sit comfortably in the shadow, and it is often used as such, while just nearby there is a large part of the street that was covered by the users to function as a place outside to get air, usable by anyone who wants to.

P10 refers to the entry point to "Cassova Alley", and it is very dense in activities related to commerce, which can be seen from the image (Figure 4).

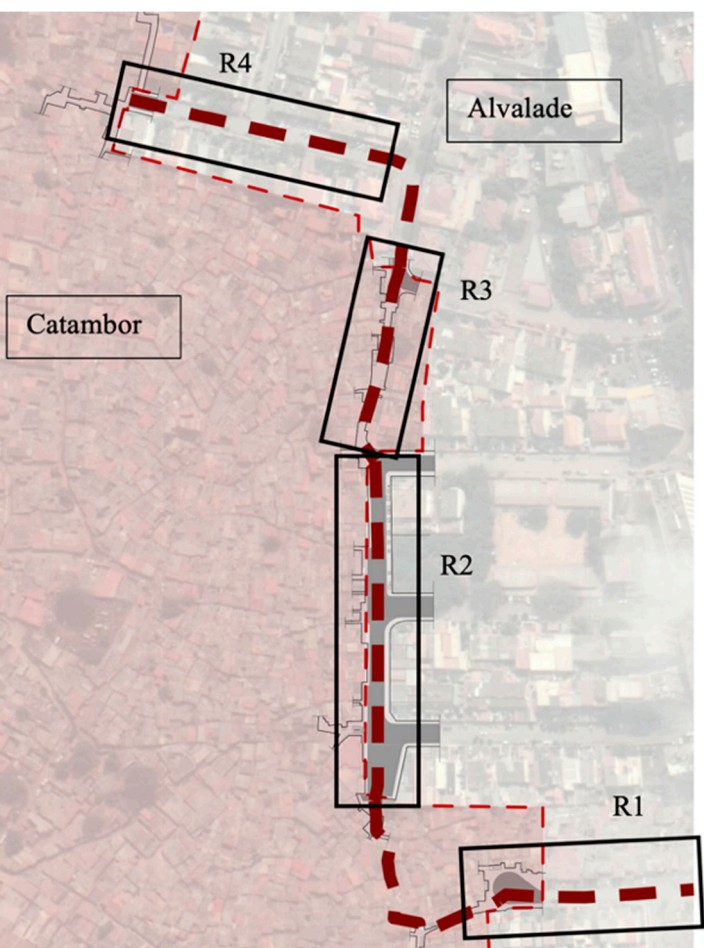

**Figure 4.** Survey Areas.

The research area is composed of a total of four different streets that border or enter Catambor in different ways, therefore characterized differently according to type, levels

of transit, and types of behavior (Figure 4). It is composed of: Dr. Egas Moniz street (R1), also known as "Rua do Notário" (Notary Street), which is the southernmost street of the research area, borders the musseque on a cul-de-sac, leading to an area with no passage of vehicles from the border point onward; P. Marien da Malta Avenue (R2), which borders Catambor at length, effectively providing a direct comparison between both neighborhoods in regard to housing and infrastructure; the "Cassova Alley" (R3), which is an extension of the P. Marien da Mata Street and connects two different streets outside the musseque itself; and, lastly, the "Elder Neto Street" (R4), which corresponds to the northernmost street in the research area and, just like, the "Egas Moniz street", also borders Catambor on a cul-de-sac. The study extends to some of the streets that enter the musseque, provided that there is a direct connection to the border streets mentioned before to provide a comparative look toward behavior and space usage.

A series of behavior settings were found, which in turn were then classified and grouped according to similarity (Figure 5).

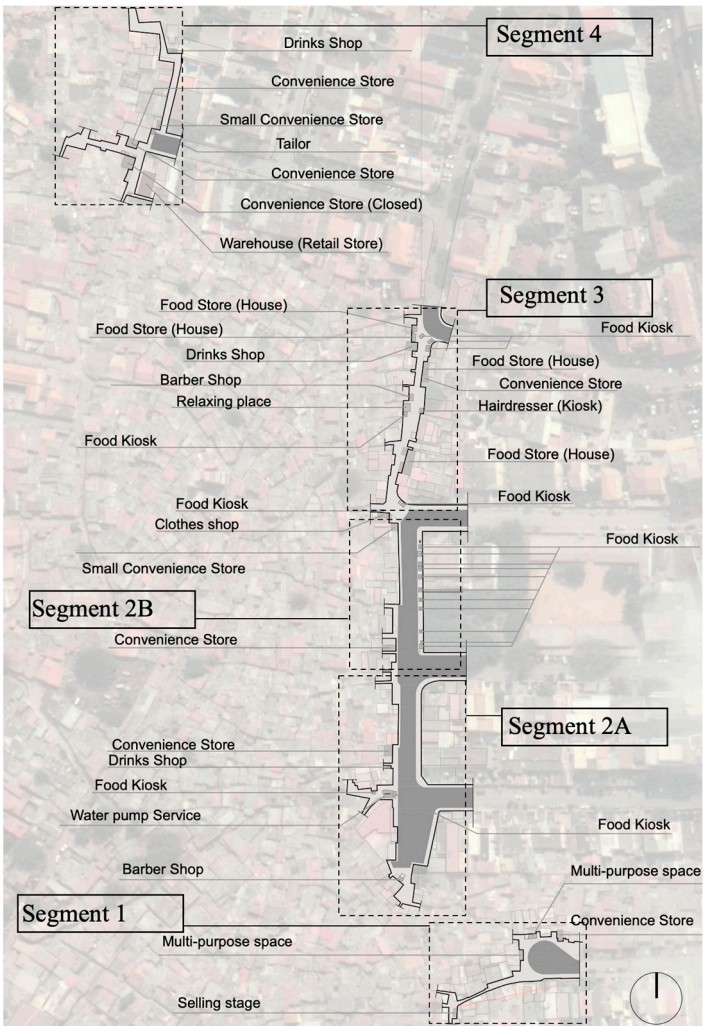

**Figure 5.** Characteristics of the Border.

As stated before, the study focused on acquiring data from the border that consists of three different streets: Elder Neto Street, corresponding to segment 4 in Figure 3; P. Marien da Malta avenue, which is represented in the figure as segments 2A and 2B; and Dr. Egas Moniz street, in the picture corresponding to segment 1, extending itself to some of the alleys that connect to those streets. For that reason, there was a need to dive inwards toward some of those streets that connect to the border to understand the nature of the

behavior inside the musseque, and in turn, conciliate the intended use cases for the streets and the influence of the uncovered behavior settings onto the inner streets.

The most noticeable of those streets is known as the "Cassova Alley", which is an extension of the P. Marien da Mata avenue, corresponding in the figure to segment 3 and making a connection to two different points outside of the musseque itself.

## 3. Results

### 3.1. Behavior Settings

A total of 17 different activities happening on the streets were uncovered, which in turn were then classified into Behavior Settings and grouped into six different categories according to similarity (Figure 6).

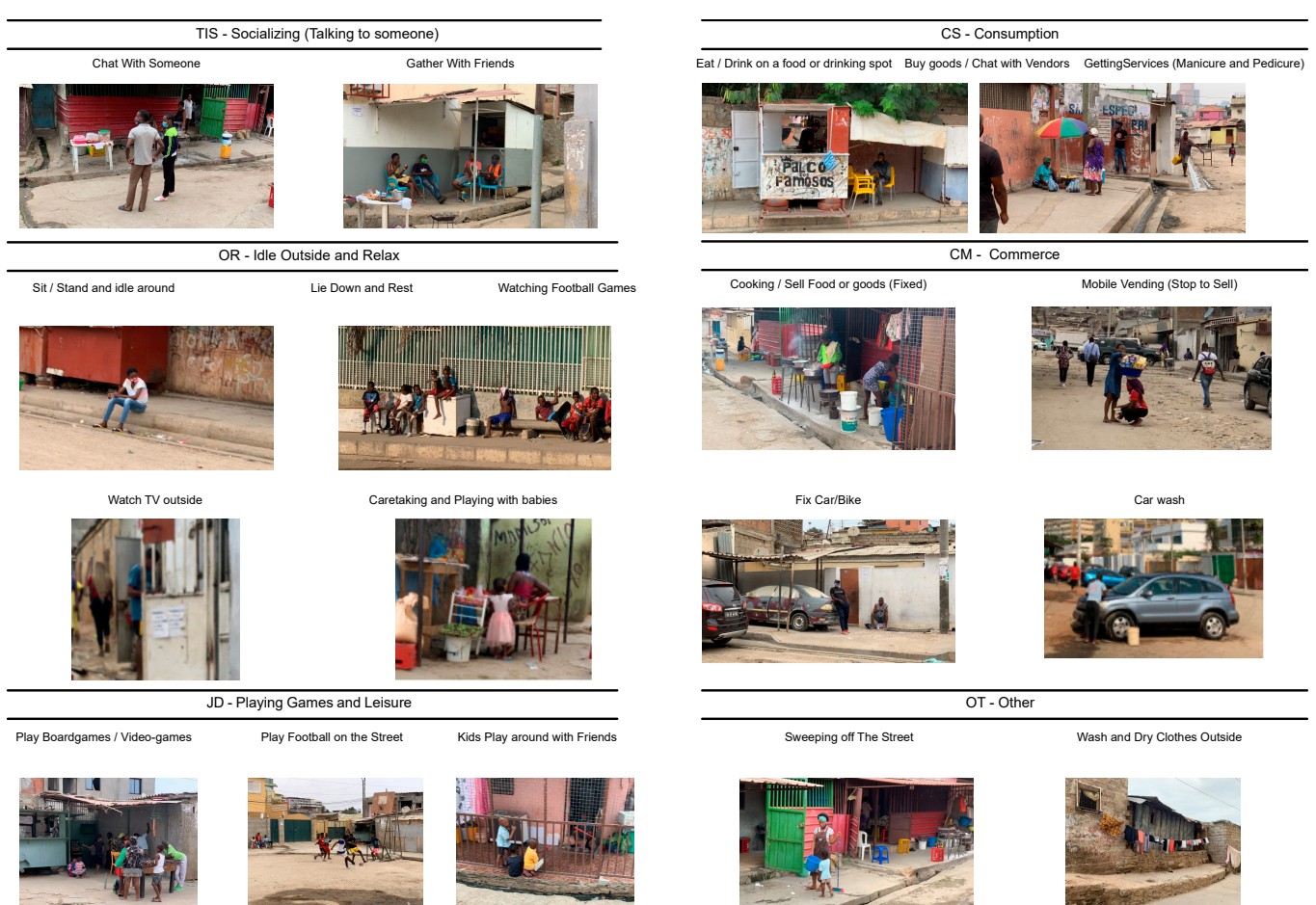

**Figure 6.** Identified Behavior at the Border.

The recorded activities were then cross-analyzed with the characteristics of the physical environment and then turned into behavior settings, as shown in Figure 6. The first category is denominated (TIS) from the Portuguese "Trocar Informações e Socializar" and consists of the activity of "Talking to somebody", communicating with someone or spending some amount of time with that person, provided that the focus is exchanging information and encompasses "Chat with Someone" and "Gather with Friends" as behavior settings. The second group is "OR", from the Portuguese "Ócio e Relaxe" and consists of activities where the focus would be to just sit or idle and encompasses the behaviors of idling around, lying down and resting, watching football games on the street, watching TV outside, and caretaking or playing with babies. Category number 3 is described as "JD", from the Portuguese "Jogos e Diversão", meaning "games and leisure", encompassing "playing board games" and "playing football on the street". The fourth category is described as

"CS", from the Portuguese term "Consumo de bens e Serviços", consisting of activities that would involve and focus on buying and or consuming goods on the street and encompasses eating and "drinking on the spot", "buying goods/chat with the vendors", and "getting services (manicure/pedicure)". The fifth category is called "CM", from "Comercio" in Portuguese, and encompasses activities whose main point is to sell goods or provide services, encompassing "cooking and selling goods/food outside" and "stop to sell" for mobile vendors. The last category is referred to as "OT", used to refer to other activities that in and of themselves have the primary focus of using the street as an extension of one's house, encompassing, "sweeping the street" and "wash and dry clothes".

*3.2. Behavior Mapping*

As referenced in Figure 7, the different areas showed various behavior settings with frequencies that varied according to the physical characteristic of the space as well as location and time of day, as the study later reveals.

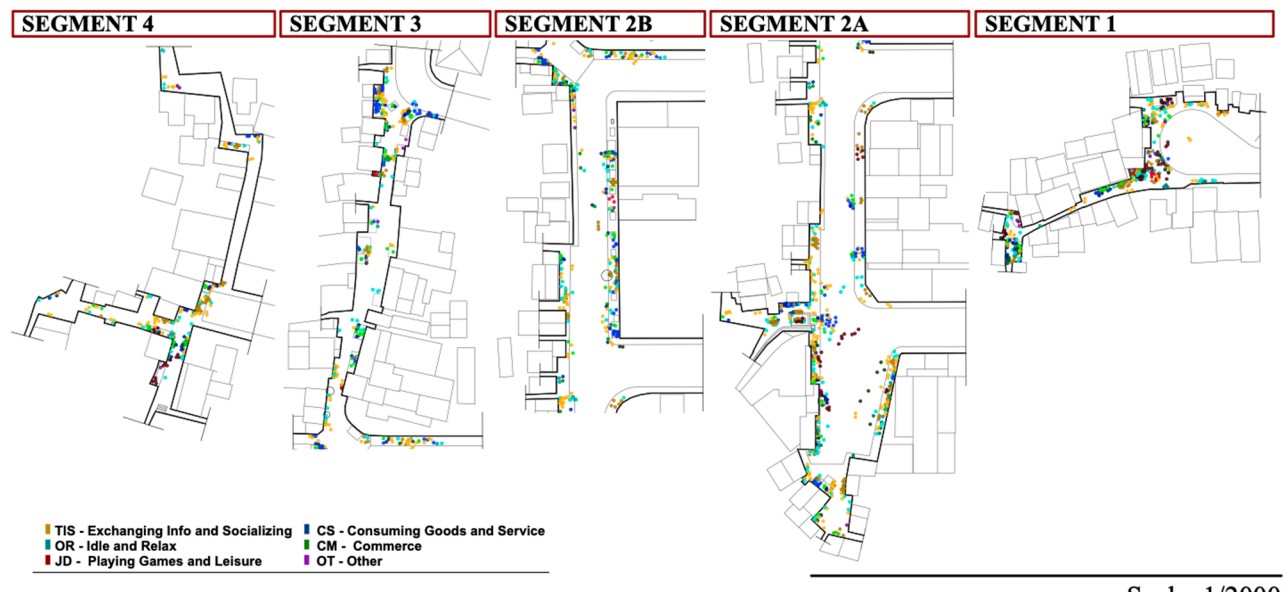

**Figure 7.** Behavior Mapping.

Segment 1 showed a high concentration of the behaviors "consumption of goods or services" (CS), socializing (TIS), and playing games (JD), which could be located in certain specific spots depending on the geometry and location. Segment 2 was the richest for commerce (CM), accounting for most of the commercial spots and most of the behavior that happened as a consequence of that, as the interviews would also corroborate. Segment 3 was the most disproportionately commercial in nature, despite the narrowness of the street itself. However, as the data suggests, there is a connection between the number of potential customers and the placement of commercial spots, and places with shade and sitting spots for consumption appear as a result.

In general, as previously stated, the behavior mapping shows a variation of the behavior depending not only on the time of the day but also on the geometry of the streets itself and the surrounding buildings, with a high concentration of commercial behavior on the border, more specifically in the wider areas such as "P. Marien da Malta Avenue" and "Dr. Egas Moniz street", seemingly due to the width of the street itself and the proximity to the city area, which allows for various business practices. On the other hand, the narrower areas accounted for most of the behavior settings related to relaxing and interactions among people, and as the data suggests, that could be linked to the narrow profile and D/H ratio, which allows for more shadows along the streets during more hours of the day. The only exception for that would be segment 3, which is primarily commercial in nature despite

the narrow profile, which seemed to be justified given its location connecting two different important spots. The P. Marien da Mata Avenue, being the easiest way of passage for people wanting to go from one side to the other, is therefore rich in the number of potential customers using the street.

Socializing (TIS) was the most identified behavior setting, mostly located in segments 1, 2, and 4 in places that were physically altered by the residents themselves by either placing benches permanently outside or by using pieces of tree trunks on the floor as benches, suggesting that the residents are willing to make changes to the place themselves to adjust to their needs. Idle and relaxation (OR) as behavior was mostly identified in segment 2 in areas that could allow for people to either sit or lie down to rest, that is, where and when there was shade and or proper conditions to sit and rest, even when alone. Playing games (JD), as a behavior, was found primarily in two different spots located in segments 1 and 2A. Those are wide areas with limited access to vehicles, allowing then for places to be used as areas to play, while many others would stay outside of the "field" watching the game. Consumption (CS) as a behavior setting was found mostly in segments 2B and 3 in areas closest to the "CM" (Commerce), and in fact, those two recorded behaviors tended to be unsurprisingly linked independently of time and place due to the nature of commerce being an act of transaction that depends on sellers and buyers. The group denominated "other" (OT) as a behavior setting tended to be a special case, and it seemed to happen mostly in segment 3 and segment 4, and it varied according to the case. In the cases of "drying clothes outside", the behavior tended to happen in areas close to one's house or immediately by the entrance door, but in areas with wide walls with no openings, providing space for putting clothes out to dry. In the case of "sweeping the streets", the behaviors could be found mostly by the entrance door, or in areas on the street where the house owners would develop a type of commercial activity, as seen in Figure 8.

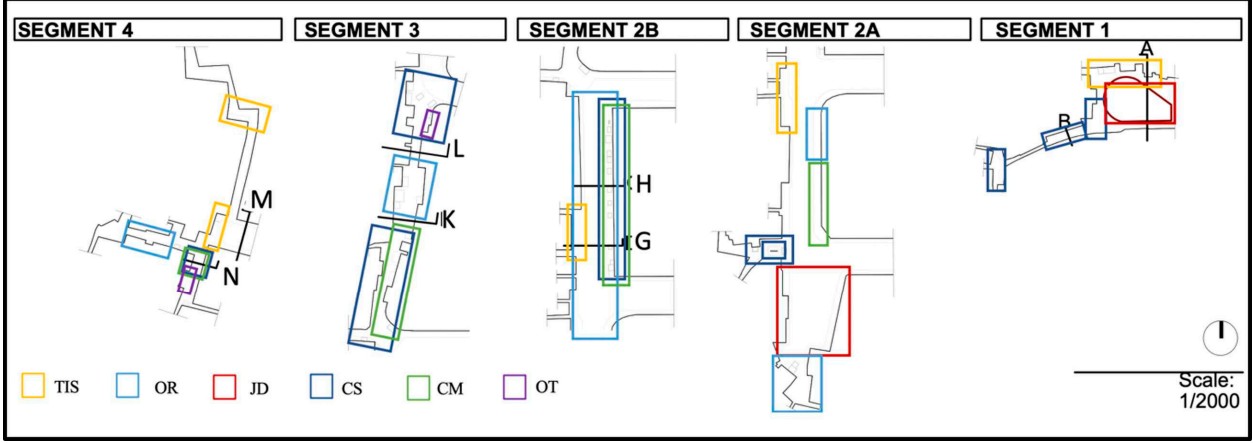

**Figure 8.** Behavior Frequency Patterns.

### 3.3. Street Usage

Sections were taken from every segment to better access the distinct types of behavior from a street usage perspective showing the variety of behaviors found in the street but also the connections and similarities of different areas regarding specific types of behavior, (Figures 9–12).

Sections A and B were taken from segment 1 in Figure 9 and represent street usage on that segment, at the border, and street usage in one of the inner streets, respectively. Although "socializing" is the most represented behavior in the section, the presence of commerce is also very noticeable, with different selling spots being assembled by the users to make use of the presence of people that come to the place as potential customers.

**SEGMENT 1**

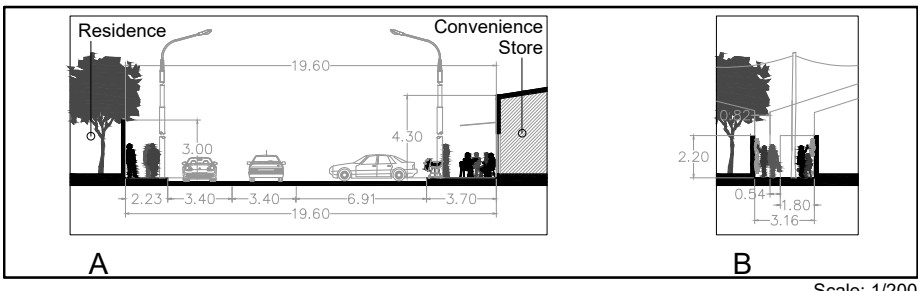

**Figure 9.** Sections A and B from segment 1.

**SEGMENT 2**

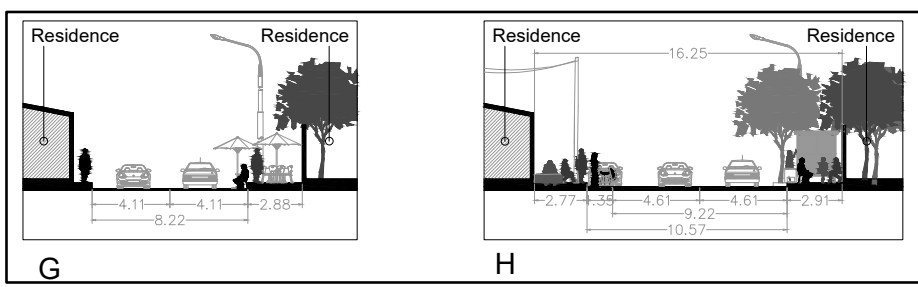

**Figure 10.** Sections G and H from segment 2.

**SEGMENT 3**

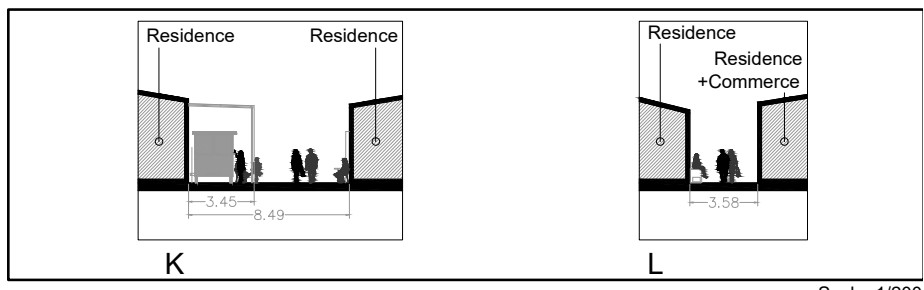

**Figure 11.** Sections K and L from segment 3.

**SEGMENT 4**

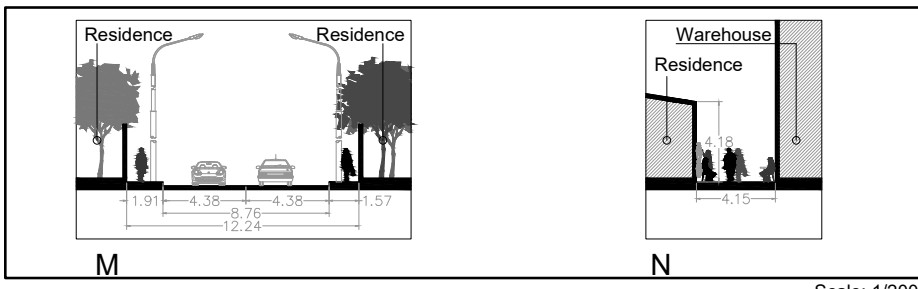

**Figure 12.** Sections M and N from segment 4.

In segment 2, sections G and H follow the same pattern, where the most represented behavior settings are consumption and commerce, and the border is used as an area to attract people as customers for the provided goods and services (Figure 10).

Sections K and L were extracted from segment 3 (Figure 11), and similar to 2B, has consumption and commerce as the most prevalent behavior settings despite not being on the border. It is frequented by users not only as a commercial hub but as an area to sit and relax as well, as pointed out by the predominance of "idling around" as a behavior.

Sections M and N, located in segment 4, are, just like A and B, a representation of the behavior at the border and inside the musseque. However, different than A and B, the usage patterns are not carried from one environment to the other; here, the differences are more noticeable, with idling, commerce, and consumption being recognized inside the musseque, with socializing and idling being the most recognizable behaviors on the Alvalade neighborhood in contrast (Figure 12).

*3.4. Time Analysis*

For a correct representation of the nature of behavior through time and for a clear idea of what influence the time of day could have in the appearance of specific behavior settings, the recorded activities were divided and characterized according to time of day and location of occurrence (Tables 3 and 4). Due to the influence that the time of day can have not only on the actions of a person but also as a complement to the physical characteristics of a place, the occurrence of behavior appeared to be heavily influenced by the time of day and position of the shade; physical characteristics of the place (with narrow streets, corner areas, and wider streets showing the different occurrence of behavior even in similar conditions); and existing equipment on the place, such as sitting areas and/or commercial spots.

**Table 3.** Behavior Distribution on Weekdays.

| | TIS 1 | TIS 2 | OR 1 | OR 2 | OR 3 | OR 4 | OR 5 | JD 1 | JD 2 | JD 3 | CS 1 | CS 2 | CS 3 | CM 1 | CM 2 | CM 3 | CM 4 | OT 1 | OT 2 | Total |
|---|---|---|---|---|---|---|---|---|---|---|---|---|---|---|---|---|---|---|---|---|
| 9:00–10:00 | 38 | 13 | 36 | 7 | 0 | 2 | 1 | 9 | 0 | 4 | 12 | 13 | 3 | 28 | 0 | 0 | 0 | 2 | 3 | |
| | 51 | | 46 | | | | | 13 | | | 28 | | | 28 | | | | 5 | | 171 |
| 12:00–13:00 | 58 | 37 | 51 | 1 | 0 | 0 | 4 | 11 | 0 | 10 | 16 | 29 | 3 | 23 | 2 | 3 | 4 | 0 | 4 | |
| | 95 | | 56 | | | | | 21 | | | 48 | | | 32 | | | | 4 | | 256 |
| 15:00–16:00 | 51 | 37 | 35 | 5 | 9 | 0 | 0 | 11 | 0 | 6 | 20 | 38 | 5 | 33 | 1 | 1 | 0 | 0 | 1 | |
| | 88 | | 49 | | | | | 17 | | | 63 | | | 35 | | | | 1 | | 253 |
| Total/Type | 147 | 87 | 122 | 13 | 9 | 2 | 5 | 31 | 0 | 20 | 48 | 80 | 11 | 84 | 3 | 4 | 4 | 2 | 8 | |
| | 234 | | 151 | | | | | 51 | | | 139 | | | 95 | | | | 10 | | 680 |

**Table 4.** Behavior Distribution on Weekends.

| | TIS 1 | TIS 2 | OR 1 | OR 2 | OR 3 | OR 4 | OR 5 | JD 1 | JD 2 | JD 3 | CS 1 | CS 2 | CS 3 | CM 1 | CM 2 | CM 3 | CM 4 | OT 1 | OT 2 | Total |
|---|---|---|---|---|---|---|---|---|---|---|---|---|---|---|---|---|---|---|---|---|
| 9:00–10:00 | 54 | 30 | 45 | 5 | 12 | 0 | 0 | 1 | 0 | 17 | 19 | 8 | 1 | 23 | 1 | 0 | 1 | 0 | 2 | 219 |
| | 84 | | 62 | | | | | 18 | | | 28 | | | 25 | | | | 2 | | |
| 12:00–13:00 | 130 | 68 | 61 | 11 | 0 | 1 | 1 | 5 | 0 | 28 | 54 | 25 | 2 | 31 | 9 | 4 | 3 | 2 | 3 | 438 |
| | 198 | | 74 | | | | | 33 | | | 81 | | | 47 | | | | 5 | | |
| 15:00–16:00 | 122 | 58 | 35 | 4 | 5 | 4 | 0 | 0 | 0 | 35 | 41 | 27 | 2 | 29 | 0 | 1 | 0 | 0 | 3 | 366 |
| | 180 | | 48 | | | | | 35 | | | 70 | | | 30 | | | | 3 | | |
| Total/Type | 306 | 156 | 141 | 20 | 17 | 5 | 1 | 6 | 0 | 80 | 114 | 60 | 5 | 83 | 10 | 5 | 4 | 2 | 8 | 1023 |
| | 462 | | 184 | | | | | 86 | | | 179 | | | 102 | | | | 10 | | |

The data collection was set up in three different time frames during the day: the first period from morning 09:00 to 10:00, the second period in the afternoon from 12:00 to 13:00, and the third and last period in late afternoon from 15:00 to 16:00.

Weekdays (Table 3) and weekends (Table 4), were divided and categorized differently due to the influences that weekday activities could have on the presence and permanence of the area and consequently the streets. Such activities would range from going to work in the morning and returning in the evening, to visiting relatives on the weekend while staying home on the weekdays, for example. In addition, weekend activities could also potentially affect behavior due to the presupposed free time that people tend to have on weekends.

Despite those early assumptions, commerce and "other" behavior (sweeping the street or washing clothes outside) remained similar between weekdays and weekends, while socializing, playing games, and consumption had considerable changes in the occurrence rate.

"Socializing" as a behavior showed the biggest jump between weekends and weekdays, with about double the occurrences being recorded on the weekends despite the relative stability of the rest of the behavior settings. This discrepancy supports the early assumption that when the residents have more free time, it is common to spend time in the street, chatting, or gathering with friends. The big jump also supports the assumption that the lack of proper electricity and poor comfort conditions at home could be linked to this rise. Playing games, idling around, and consumption followed as well, with a relative increase from weekday to weekend. Those behavior settings tended to be all directly affected by the number of people on the street, as is the case with consumption and/or the number of people with free time for recreational activities such as those registered under the playing games umbrella, with "idling around" being seemingly correlated with "socializing" as a behavior.

In addition to the variations based on the time and physical characteristics, behavior in the different segments of the research area also varied from place to place, depending on occupation trends and the presence or absence of certain causative behavior.

In segment 1 of the research area, despite socializing and commerce being the most represented activities during most periods of the day and having accounted for many of the behaviors spread across the specific areas (Figure 13), "socializing" there had a lower level of repetition of the same activity across the same place as most of the other different activities, which was perhaps due to the non-permanent nature of the behavior. However, despite that, there was still a pattern to be recognized, as most of "socializing" was recorded in the second period, that is, from 12:00 to 16:00 covering both the second and third periods, seemingly because there were simply more people on the street at those times. Commerce was the most present behavior, and mostly at all periods, both on weekdays and weekends, with distinctions in regard to the commerce of tomatoes, vegetables, and other basic goods such as being present during all periods and the food spots being assembled for the most part exclusively during noon, at lunchtime, due to the fact that on this section, most of the food spots are not fixed as it is common for vendors at segment 1 to prepare the food and cook just by the road directly to passersby. On weekends, the major difference to weekdays was the presence of a much higher prevalence of "playing games" in the form of football games in the early hours of the day.

Divided into two, segment 2 had a high predominance of consumption and commerce as both behaviors often go hand in hand, as seen in Figure 13, but the variation of behavior settings was the highest among all of the sections, particularly in the sub-section 2A, also with a strong presence of "socializing" and "idling around", which tended to intensify at late afternoon in the last period, which coincided with the appearance of "playing games" as a behavior, in both weekdays and weekends, divided between football and basketball games, by different groups of people, suggesting that street is indeed regarded as "the sports court" among its users. 2B, on the other hand, showed very little changes over time and remained mostly stable and with a somewhat established routine, with commerce being present from the first to the last period, followed by consumption, which though occasionally starting in the morning, was most prevalent from 12:00 and until 16:00 at the

end of the last period. With the one change, which seemed to be true mostly to segment 2, was that on the warmer days, regardless of the day of the week, the predominance of people seemed to shift increasingly toward the west from afternoon onward, while most of the activity under shadows remained intact.

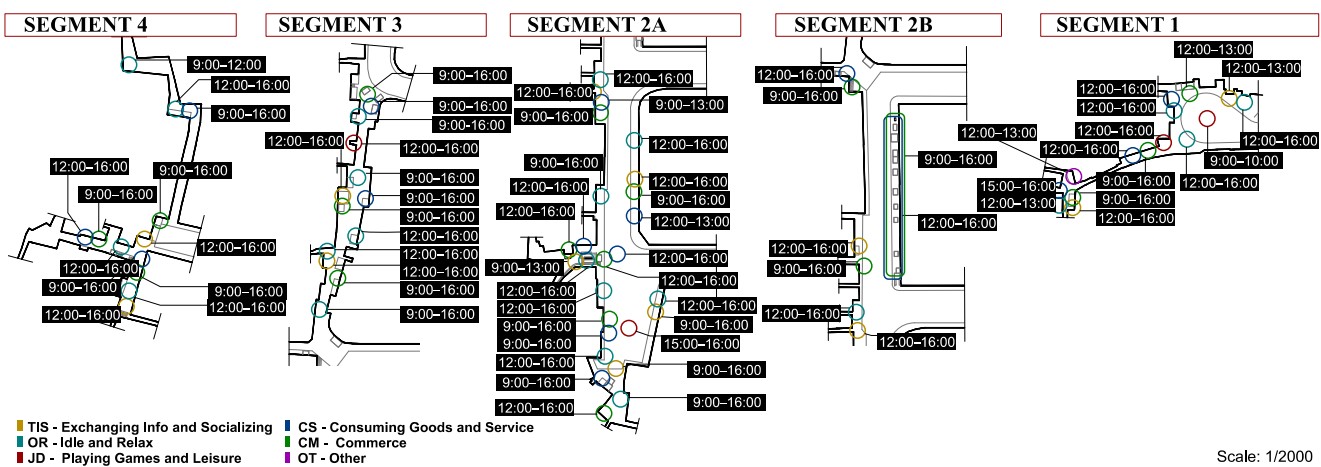

**Figure 13.** Behavior through time per space.

In segment 3, just like segment 2, commerce and consumption are the most represented behavior settings, as uncovered in the space analysis. On the other hand, different than in segment 2, the differences in behavior through time during daytime show very little differences, with most behavior settings starting, as is the case with consumption, commerce, and "idling around" being significantly represented from 9:00 and go up until 16:00, with TIS being the least represented in early morning. The variation of behavior from one place to another could be seen in some places, as some activities happened in certain areas more than others, such as "socializing" and "idling around", which could be assumed to be due, partially, to the D/H ratio on the alley, which can be measured at about 2:1 in some places and about 1 in some other places, and that partnered with shade, established by the residents themselves and the irregular geometries, led to an environment, in certain portions of the day, that was shaded at different spots.

With a predominance of the behaviors of idling around, commerce, and consumption, segment 4 showed few changes in behavior occurrence through time. Showing the same trend as all the other segments, commerce was recorded with frequency from 9:00 to 16:00 and consumption started to show frequency from noon, with people starting to come outside to buy ingredients for the house just before lunchtime onward. Socializing did show differences in a prevalent location, with groups of friends being spotted inward in the slum with less frequency at the later hours of the day, while the same activity was spotted with more frequency at the exact border with more frequency as time progressed, which can be linked to the increase of "boredom" the more you stay at home, as well as the increasing shadows in the street as soon as the sun leaves its zenith, leading to a more comfortable place on the streets due to the increase of shadows in the street.

### 3.5. Summary of the Behavior Analysis at the Research Area

In general, the time analysis confirmed the predominance of "socializing" as the most prominent behavior in the area across all times, but more than that, the disproportional increase of the same type of behavior on weekends, with the largest increase during each day being registered from period 1 (09:00–10:00) to period 2 (12:00–13:00), whereas period 3 (15:00–16:00) remained relatively similar to the previous one, with very minimal changes (Figure 14). The analysis further indicated that "idling", consumption, and commerce were the following most common behavior settings, with both idling and consumption showing a small increase from weekdays to weekends and commerce showing a marginal change in

the same period. The daily changes for all three behavior settings were similar, peaking in occurrence rate in the second period but falling slightly in the third one. Similar to socializing, playing games was the only other behavior setting with a considerable increase between weekdays and weekends, and again, just like socializing, the rate of occurrence increased from period 1 to 2 but remained relatively similar from period 2 to period 3.

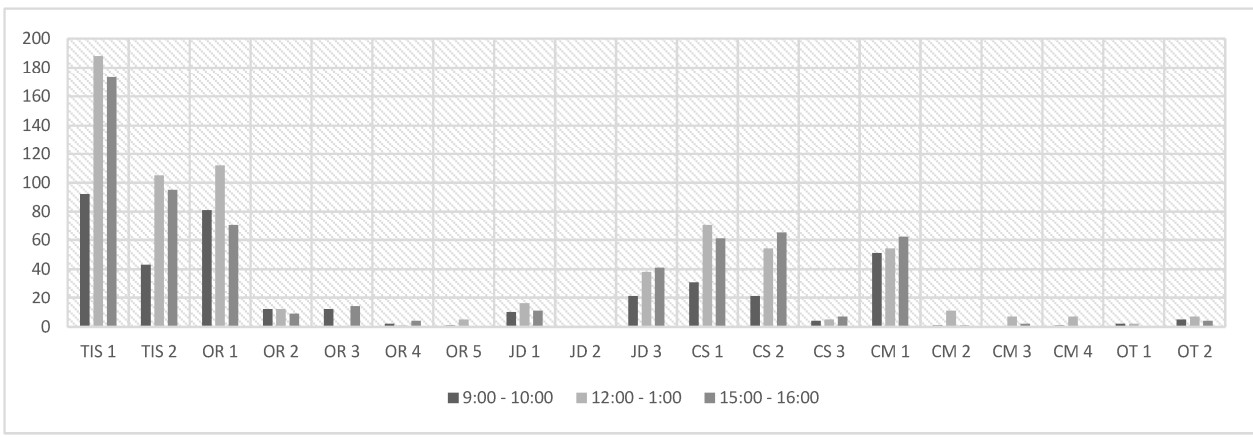

**Figure 14.** Behavior Distribution by Time of Day.

Weekends did see a predictable increase in the occurrence of the behavior, with recreation and/or idleness-related behavior taking the highest spot, followed by commerce and consumption with a mild increase as well. Period 2 saw a sizeable increase from period 1 in most behavior settings, with period 3 largely keeping the same rate of occurrence as period 2.

In addition, the results point toward the occurrence of the multiple types of behavior settings being connected to different factors such as the structure of the physical environment, the closeness to the metropolitan area of the city, as well as factors originating from the livelihood, such as boredom at the house, as uncovered by the interviews. That is particularly true regarding socializing and idling around as a behavior, as very often the observation would point to the fact that the same people and the same groups of friends would be using the spots to sit and talk, as well as the same users using the same spot to idle and spend time, repeatedly around the same timeframe on different days. Furthermore, the result from that repeated behavior would be the addition of elements on the street to adjust it to those activities, such as the broken fridges found on the streets, used for people to both sit on and put products on, by vendors, as well as the wooden trunks on the sidewalks to function as places to sit, since it was common for the users to spend longer periods of time sitting, idling, or chatting on the spot.

There is a connection that can be established between the proximity to "Alvalade" and the occurrence of commerce and consumption behaviors, which in turn resulted in several permanent and semi-permanent changes on the streets, such as the occupation of the sidewalks leading people to using the road as a passageway instead of the sidewalk on many occasions, while the sidewalk became a zone to sit down, eat, and mingle. Those factors, put together with the analysis of the alteration of the public space, led to the conclusion that those alterations on the public common space and the interaction between spaces can be summarized and divided into three different categories Figure 15 1—semi-permanent alteration on semi-public space, such as the trunks being placed in front of the houses for socializing and idling behavior or the front of the houses being used as a commercial spot to cook and sell food; 2—semi-permanent alteration on public space, with the occupation of the sidewalks with trunks and broken fridges as places to sit and watch the game; and 3—permanent alteration on public space, as represented by the complete occupation of the sidewalks by the "roullottes" and food kiosks in segment 2.

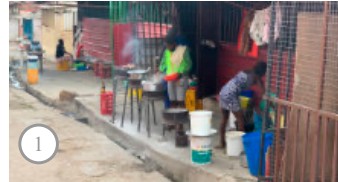
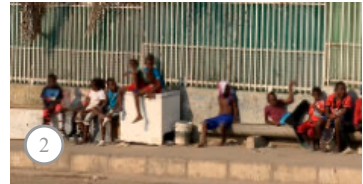
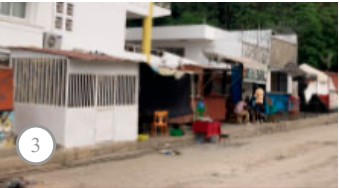

**Figure 15.** Categories of Space usage and alterations on the Physical Environment.

*3.6. The Environmental Behavior in the Eyes of the Users*

The final part of the study consisted of several interviews conducted in the different segments of the research area (Figure 16). A total sample of 61 participants was interviewed to make the connection between the findings reached by the behavior mapping of the area and the reasoning behind their happening in the eyes of the users, in order to understand what is at the basis of the decision-making process in the minds of the users.

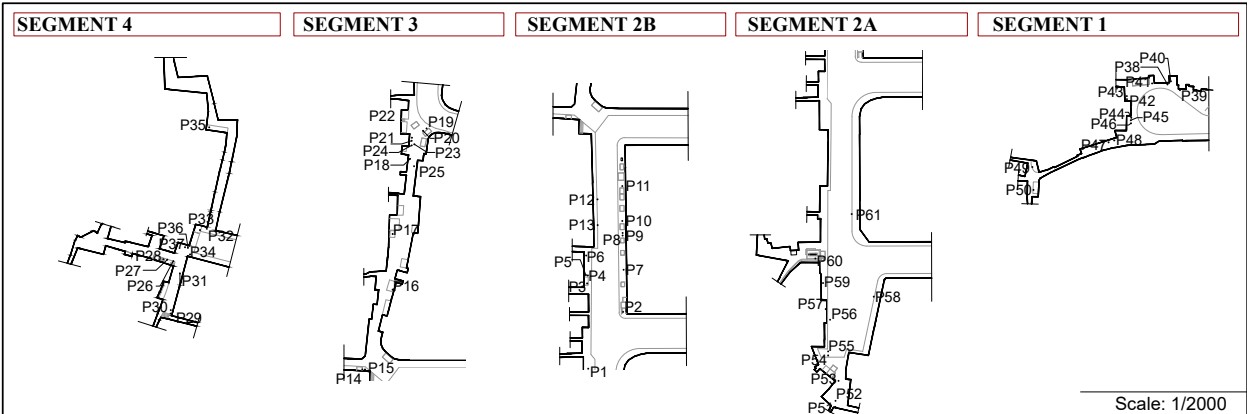

**Figure 16.** Mapping of Interview Respondents.

By looking at the most common types of behavior by user, repetitions, preferences, and impressions of the distinct characteristics of the environment, the study managed to give a clearer image of how the different sets of behavior come to be, and which types of responses manifested in the space from the viewpoint of the users themselves.

A total of 13 interviews were collected in segment 1, and while the most recognized behavior was socializing and commerce, as mentioned before in the interview results, the major "activities at the moment" were found to be "eat/ drink on a food or drinking spot (CS 1) and "idle around" (OR 1), and the potential reason for that is because people talking to others were noticeably less prone to giving the interview.

In segment 2, the results in the 2B section showed a difference when it comes to the respondents and their activity at that time, with "cooking and selling food and goods" (CM 1) and "gather with friends" (TIS 2) being the most noticeable among the various behavior patterns.

However, just as the previous data showed, according to the respondents themselves, the most common usual behavior was "socializing" TIS 1, often followed by "idle around" (OR 1), as many people showed an inclination toward staying outside instead of inside and often to escape the boredom inside the house, as the interview results will also point.

The responses in segment 3 differed fundamentally from the other segments, but that can be primarily attributed to the fact that the segment is much more commercial in nature.

Furthermore, in accordance with the previous data on the analysis, most of the responses point to cooking/selling food (CM 1) and eating/drinking at a food spot (CS 1) being the most present activities, further solidifying the notion that segment 3 is primarily a commercial hub, with the exchange of services being the primary focus, being it selling goods or consuming them.

In segment 4, with a total of 12 respondents, chatting with someone (TIS 1) and gathering with friends (TIS2) as well as "idling around" (OR 1) were the most common behavior patterns by the time of the interview.

On the other hand, when it comes to the usual activities on the street per person, alongside socializing (TIS 1), eating/drinking at a food spot (CS 1), and cooking/selling food (CM1) were the most common behavior patterns, which goes more in line with the data from the observation stage.

### 3.6.1. Sample Structure

The sample was composed mostly of males, and though the most represented age group on a five-year age group distribution was the one of 25 to 30 years old, on the same type of age group distribution there were no significant differences between all of the age groups between 20 to 45 years old, but of the respondents among all of the age groups, on a twenty-year group distribution, the vast majority of the users on the sample were grouped between the ages of 20 and 40, representing an age group of primarily labor force aged adults, that used the streets for the most varied types of activities, whether being labor-related or not (Figure 17).

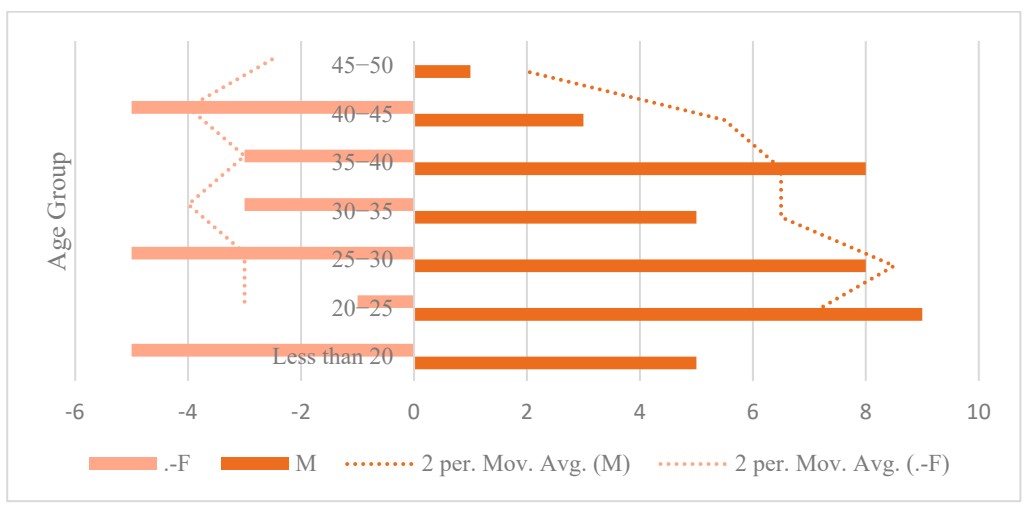

**Figure 17.** Age Group and Sex.

As comments from some of the respondents would later support, the larger presence of males on the street could be linked to cultural implications, with men being often perceived as the providers for the family and putting them on the streets more often, the notion that women who spend too much time on the street can be looked at as "of a life without virtue", or ultimately, that the streets are safer for men than they are for women.

### 3.6.2. Reasons behind Behavior and Impressions Regarding the Environment

Both previous questions looked at the behavior occurrence of the behavior in the eyes of the users, but the question of "reasons for using a certain spot" attempted to characterize what the reasons for the occurrence of behavior are, and that started by looking at the reasons for the choosing of the particular area to stay, in order to understand what the respondents found useful or of meaning when it came to the physical space (Figure 18).

For that reason, the question was framed to take different types of answer from various respondents to allow for a qualitative assessment of the data and allow for the collection of all the possible reasons that the users could think of, which resulted on the data being coded and measured by the number of mentions per reason.

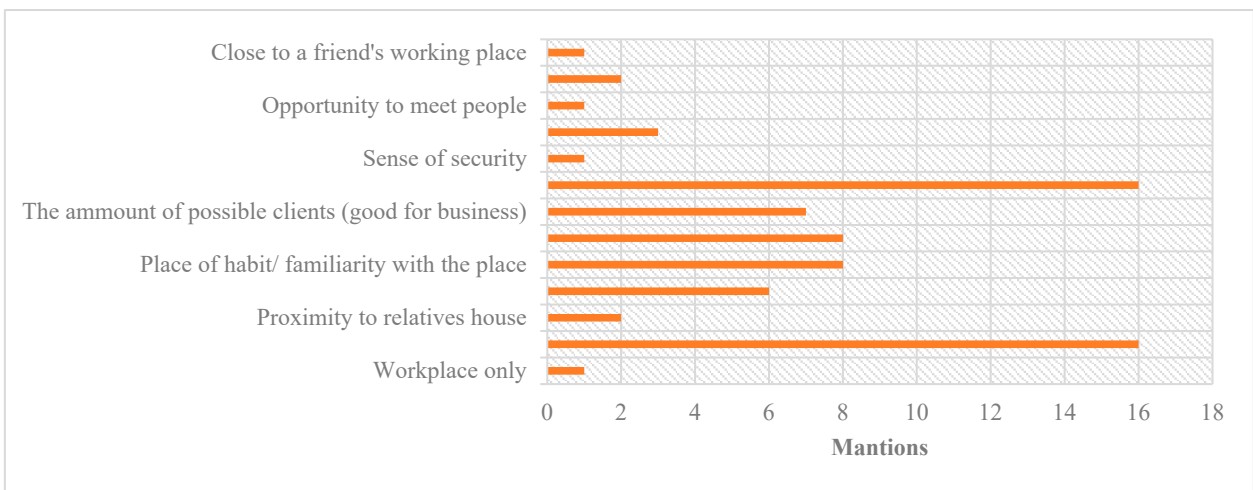

**Figure 18.** Reasons for Using the specific spots.

From the answers, it can be perceived that the most important element for the respondents was a certain proximity to the house and social features of the place, which from the responses can be namely pointed to as "the lively environment", "hospitality", "degree of Tranquility", "sense of familiarity", and for being the place "where the friends are". After that, other reasons also came into place, such as the built habit, to go to the same place as well as the level of comfort (air flow, benches, shade, and wideness of the space itself). Pointing to the existence of a higher regard to the social features of the environment by the users in comparison with physical features of the place.

While exploring the questions regarding the individual impressions of the environment, it was still necessary to dive into what the users valued throughout the physical aspect of the area itself, with "No opinion" accounting for the most the mentions, indicating a certain difficulty of the users to perceive positive points on the streets. However, several people tended to opt toward "The Commercial spots" next to some places as reasons for the visit when it comes to the physical space, as seen in Figure 19. This supports the findings from the behavior analysis, but other responses pointed toward a sense of beauty in the space noticed by some users, the openness and wideness of certain spaces, and other reasons like the proximity to the city were laid as main points of attraction regarding the physical environment.

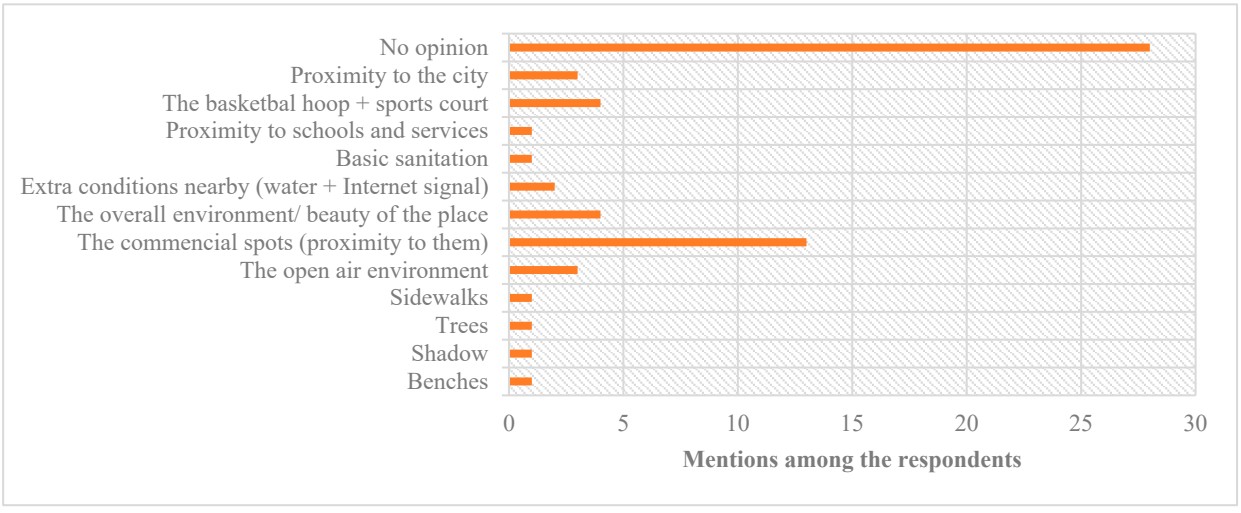

**Figure 19.** Attractive Points based on the Physical Aspect.

On the other hand, regarding points of improvements, it was also difficult for many respondents to point out, as many respondents could not notice or think about any similar to the opposite question. This points toward a degree of complacency for either the positive or negative aspects of a space by the users. However, among the responses, sewage treatment was the main point of improvement to be pointed, followed by the infrastructure in general, which can be characterized as the buildings and state of the road, hygiene of the place, lighting at night, and the tight urban layout, and the data was mostly fairly distributed among the different segments of the survey area. Furthermore, when broken down into the different segments, despite the irregular layout of segment 3, most of the respondents were less prone to identify points of improvement, with most respondents claiming that they could not perceive any possible point for improvement (Figure 20).

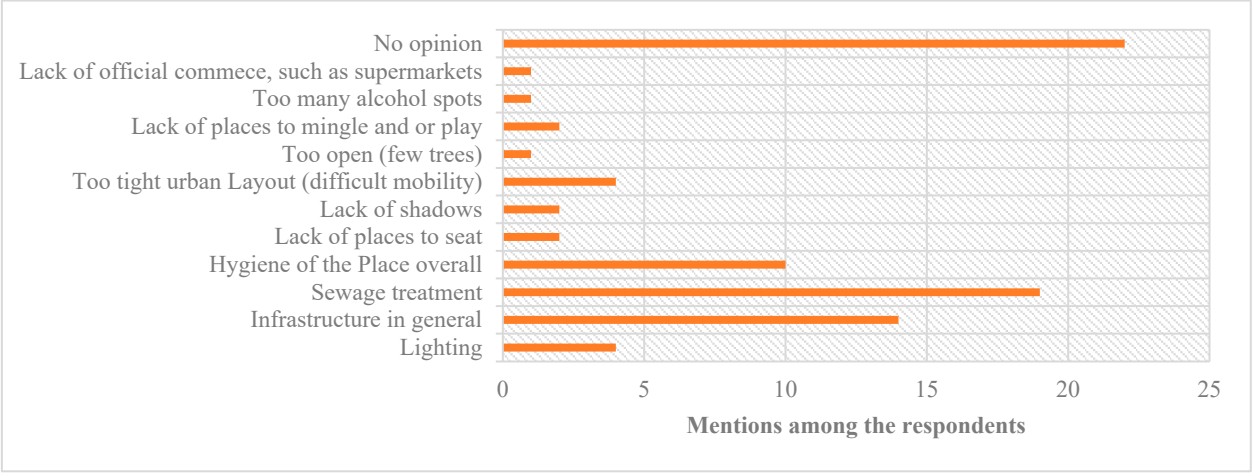

**Figure 20.** Points of Improvement based on the Physical Aspect.

Alongside the physical aspect, the non-physical was also analyzed, and for the question of attractive points, the friendly environment accounted for most of the mentions, followed by a certain sense of tranquility by the neighbors and the lively environment granted by the commercial spots and the variety of people with diverse backgrounds sharing the same places. However, a sense of community was also part of the mentions alongside the familiarity of the place to the users themselves, as seen in Figure 21.

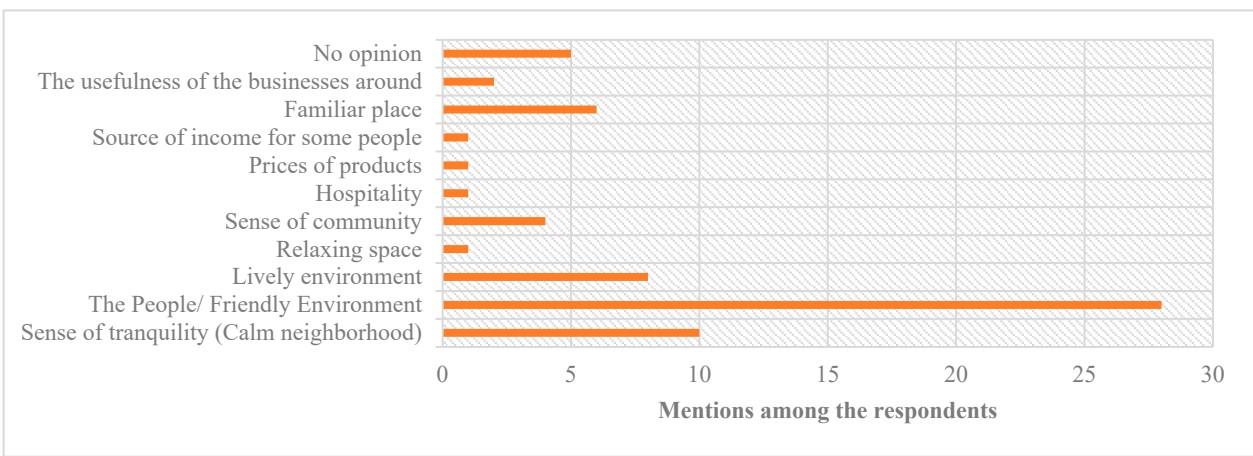

**Figure 21.** Attractive Points Outside of the Physical Aspect.

The last question of the interviews referred to the different points of improvement outside of the physical environment. Although "No opinion" accounted for most of the mentions by the respondents, suggesting a certain degree of comfort by the users,

"delinquency levels" and a "high level of alcoholism" were the most present from the mentions from all the data, which, according to the responses, seemed to be connected to the number of drinking spots in the different areas (Figure 22).

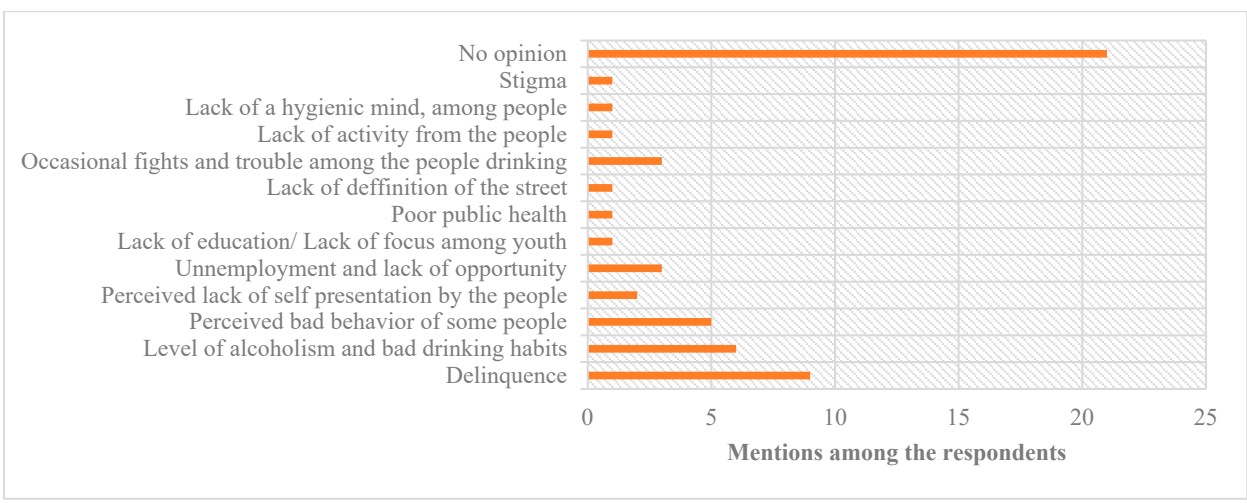

**Figure 22.** Points of Improvement Outside of the Physical Aspect.

*3.7. Summary of the Interview Results*

The interviews allowed for a deeper assessment of the characteristics of the behavior in the research area, from the viewpoint of what is causing it; what the implications are from it; and, lastly, what types of responses did the environment trigger regarding the users.

The results seemed to support the early findings in regarding the types of occupation of the users of the street in ways that both results ultimately suggested that for many, those streets can be both a place to relax and mingle, and a place to look for earnings all at once. When that fact is partnered with the recurrent problems caused by the irregular design, such as the lack of stores for basic goods or public spaces for self-amusement, it creates the conditions for the occurrence of some of the behavior settings such as commerce and consumption as suggested by the behavior analysis.

On the other hand, the data also suggests that some of the behavior settings, such as socializing and idling on the streets, have a direct correlation with the perception of the users toward their households, which is confirmed by the fact that many of the users considered boredom inside the house as one of the main reasons for "escaping" to the streets.

Those reasons partnered with the proximity of the street to "Alvalade" next to the core of the city, and the lower prices of products offered in the zone in comparison to other places to eat or drink increase the number of users from the outside, which increases consumption, in turn increasing commerce as a behavior and ultimately increasing the popularity of the place, therefore, inviting even more buyers, consumers, and bystanders to the place.

Lastly, it became clear from the investigation that for many of the users, the commercial environment of the areas was one of the main points of interest regarding the physical space, and in turn, the state of sewage treatment was the main source of concern for the users. On the other hand, there was some awareness of the problems of the neighborhood, and while the delinquency levels inside the neighborhood were the main point of concern, the environment caused by the people sharing the same spaces in the morning and the friendly nature of the people were suggested to be the main points of interest for the same users, giving weight to the connections built and the value of community from the eyes of the users, built from the improved street environment itself.

## 4. Discussion

As pointed out by the literature, the grouping of people with similar social and economic characteristics happens through the pursuit of urban resources, which can be land but also hospitals, public buildings, and other amenities that stem from living at or near the city (Park and Burges, 1925). Moreover, as observed with the Brazilian "favelas", and similar to shantytowns around the world [34], the paradigm brought by slums is that such extremely different realities with different sets of behavior do eventually create a disconnect between the populations on both sides of the border, generating stigma against either side. However, the behavioral exchanges happening at the border do not only contribute to the building of a sense of community from both groups, as suggested by the results, but also shape opinions and reduce the stigma that comes from the "musseque" itself.

The study comes from the necessity to look at the slum and city environment not as opposites but as two environments with both a proximity relationship and an extreme gap between them, and to the necessity to bridge that gap through the street environment.

The problem lies in the inadaptability of the border streets to account for the uncovered behavior settings, which, as the hypothesis pointed out, could be a key factor for the perceived misuse of the streets itself, as noticed from the existence of selling spots in areas that would otherwise be destined for circulation, the continuous execution of different types of stationary behavior on the middle of the road as the section of uncovered behavior discusses further, and on that note, the analysis was focused on trying to solve the following questions:

- Which urban design changes or adjustments happen as a result of human interactions at the place?
- How that behavior manifests itself, what causes it, and what consequences come out of it?
- What consequences does the unplanned design have on the area that directly affect the slum dwellers?
- What would the role of those human interactions and community building as a whole be?

The research confirmed the hypothesis that the proximity of both neighborhoods is one of the key factors for the occurrence and permanence of most of the behavior found in the border's streets, with "commerce" being the most directly affected. Other factors such as the size of the street and the presence of shade were also influential overall. The absence of amenities seems to have guided the users toward the largest streets with the highest potential frequency of customers paving the way for the informal, yet practical, use cases assigned to the area, which in turn give rise to other behaviors such as "socializing", "idle", and "playing", essentially bringing life to the streets and a sense of warmth perceived positively by the users.

Despite the common view of disdain or contempt, given its informal nature, the results suggest that the previously mentioned production of behavior and the practical use of space in the area are more than often borne out of necessity, as supported by the findings. From the wooden trunks laid on the street to create sitting areas to the selling spots built on the sidewalk, altering it permanently, the self-made design changes give strength to the "unity theory" of space conceptualized by Lefebvre [35]. This points toward social space being composed of the physical (conceived space), the mental (perceived space), and the social (lived space) [36], transcending the idea of formal or informal, and raising the questions of 1—"what good is the designed space if it does not respond to the needs of the users" and 2—"how evil can the production of informal space be, if it breaches the gap between the conceived space and the sense of satisfaction by the users".

The findings point to confirmation of the hypothesis that small adjustments do occur on the streets conducted by the users themselves, both on a conscious and an unconscious level, to compensate for the lack of adaptability that the streets offer to accommodate the behaviors existing in the musseque that are fundamentally unique.

The identified differences in behavior can be intrinsically connected to cultural shifts within the population, and the results show that the way the musseque residents behave regarding public space is connected to their needs and manifested either as a response to the lack of services or as a direct adjustment of the environment to provide the upmost perceived comfort. The "trunks on the sidewalk" and "broken fridges" transformed into high benches, behaving as a replacement for places to sit and chat and mingle, serve as an example. Although sociocultural needs, such as the need to mingle out in the street, are potentially connected to the levels of unemployment and lack of stable electricity, guiding people to be outside as a form of "escaping boredom", it is still a natural development of the culture in the musseque, making it a problem to planners to deal with. The streets serve as an example of the influence of the culture of the people in the place they live in, but also work as a reminder of the inadequacies of the current state of planning toward the musseque residents.

From the behavior occurrence patterns and looking at the changes on the street, the implication is that the more people get the chance to stay at home (that is, when not going to work or simply leaving the neighborhood), the more the users' default to "meeting", "gathering", or simply heading out to "chat on the street". This suggests that potentially, looking at the interventions the users conduct on the street, such as creating places to sell and sit, to either consume or mingle, gives room for the hypothesis that for the needs of the Catambor dwellers, in the ability to give and receive services or simply the "connection" between the people, users, vendors, or friends, and the "pass and stop by" behavior, has a higher degree of importance that the initial function of the street is to provide a connection between different spots.

From the current existing plans to deal with "musseques" in the city of Luanda, it becomes clear that the focus is to solve the problem of slums through the creation of newer neighborhoods that better suit the view of the city. This poses a problem when the comparison between the size of the intervention and the problems with relocation are put on the table.

From the findings, an improvement method focusing on the improvement of the livelihood in the slum, which would benefit not only the musseque but the city side as well, seems to be the most plausible solution. Focusing on the understanding and implementation of the United Nation's SDG 11 (sustainable cities and communities) in its efforts, such an approach shall consist of the complete characterization of the behavior settings to propose punctual action and steps based on the needs recognized at the place to improve livelihood and improve the level of response of the streets to the user's needs, reduce the level of individual alteration to the common space made by the users of the common space as a response to the lack of services and lack of essential amenities at the slum, and functioning as a base for:

- Improving the street environment;
- Providing extra services to the users;
- Community building;
- Improve the vending environment for vendors.

Further research is needed to further characterize the common space areas both inside and outside of the slum, as well as compare the different slums in the city of Luanda and explore the variations in behavior to establish a proper framework of action for architects and other technicians, and further optimize studies for researchers and scholars. In addition, expanding the scope of the research to encompass the reality of different countries, from Africa, South America, and Southeast Asia, would give it a much broader degree of relevance to different realities with a similar problem.

## 5. Conclusions

The behavior at the border is intrinsically connected to cultural shifts within the population, showing that the way the musseque residents behave in regard to public space is connected to the musseque residents, their needs, and the services provided as a response

to the lack of services or the changes on the environment, with the trunks on the sidewalk and broken fridges transformed into high benches behaving as a replacement for places to sit and chat and mingle.

The research has found that:

- Small adjustments occur to adjust behavior and lack of services;
- Geometric irregularities and unused spaces often get used as either a selling spot or as a place that young folks repeatedly use to gather and mingle;
- There seems to be a level of acceptance by the people of the appropriation of space;
- Community is built both as a natural development of culture and as a response to life circumstances, such as the low employment rate, which influences the decision to "go outside and mingle" as the data points out;
- The commercial spots guide a big part of the interaction and the behavior settings that happen on all the spots, independently of the level of occurrence found at the different sections of the research area.

However, in regard to the users of the space themselves, there is an assumption that comes from the slum dwellers that the space is entirely common but different than the notion that the space belongs to nobody; therefore, everybody can use it, and it is a notion that the space belongs to everybody that needs and wants to use it, and because of that there is an acceptance of the people of the appropriation of public space, suggesting a strong sense of community.

The other side of what the results show comes with the fact that contrary to the hypothesis that the stigma from the outside could potentially stop people from using the border streets, the results point to just the opposite; the border between the two neighborhoods functions as a bridge between those two realities, effectively building a third community in the middle, based on the exchange of services.

**Author Contributions:** Conceptualization, Y.O.; methodology, Y.O., S.M. and R.N.; software, Y.O.; validation, S.M. and R.N.; formal analysis, Y.O.; investigation, Y.O.; resources, Y.O.; data curation, Y.O.; writing—original draft preparation, Y.O.; writing—review and editing, Y.O.; visualization, Y.O.; supervision, S.M.; project administration, Y.O. All authors have read and agreed to the published version of the manuscript.

**Funding:** This research received no external funding.

**Institutional Review Board Statement:** Ethical review and approval were waived for this study, since no data and information related to the ethical guidelines were at the discretion of the committee at Hokkaido University.

**Informed Consent Statement:** Informed consent was obtained from all subjects involved in the study.

**Data Availability Statement:** The data regarding the findings in this study can be found at: https://drive.google.com/drive/folders/1EccFFm80lopa13r6n70z66mKIw0iUbFS?usp=sharing (accessed on 23 December 2022).

**Conflicts of Interest:** The authors declare no conflict of interest.

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
