# Peer review of "Common Space and Behavior at the Border between Slum and Metropolitan Area: The Case of “Catambor” and “Alvalade”"

_sustainability, doi:10.3390/su15065293_

Round 1
Reviewer 1 Report
The work seems interesting, entailed a lot of hard work, has some good data and I would ultimately like to see this work in publication. However, it has some clear issues that need to be addressed first, and I am not entirely convinced that this is the right venue for publication.
1. What is the research question? There are points that suggest things like purpose, significance, and applications, but I was never clear with a clear and exact statement of the question the research is trying to answer. (23:812) mentions a hypothesis, but this is the first point at which it is mentioned in these clear terms.
There are two further related points that could help with the resolution of this issue:
a. The need to connect the musseque to the neighborhood is not made clear.
b. What is the connection to sustainability? Mention is made of SDG11, but the connection to the research is never made explicit as to how this research offers guidance in this direction. I can imagine connections but they need to be made clear and explicit, with implicit links seen throughout. I would suggest discussing SDG 11 and its component parts at the start, and them connect the parts of the study, organically, explicitly, and implicitly to the goal and its sub-goals.
2. Why was Catambor selected as the location? 5:204 says, “And after deep analysis…” What analysis? What was the basis for the selection? Perhaps you could make more explicit the connection in 6:231-235 in the earlier discussion?
3. The numerous codes for the various behaviors become confusing. Either provide a table for the codes or use plain language descriptors in the text (with or without the accompanying code).
4. Figures 21, 22, and 24 all had “None Perceived” as by far the most popular choice. There needs to be more accounting for this in the text. The figures could be omitted, with mention of the apparent anomaly. Alternately, the results could be explained with a theoretical basis as to why the result is unobjectionable, or as to how and why the result is meaningful.
5. The Discussion lacks a clear summary of the results. I hesitate to suggest more tables or figures, but that would be one way to streamline the presentation. Alternately, there could be a first paragraph summarizing the results. This might seem redundant following the summary of the interview results, but a single place that synthesizes the interviews and observations would clarify the results substantially.
6. There are frequent English errors, some requiring a close reading for the correct understanding of the passage, in addition to numerous errors and typos (e.g. 4:182 “Rad” for “red”; 3:119 “pursue” for “pursuit”). There is a systematic issue with miscapitalization, with as well as frequent use of conjunctions at the beginning of sentences, giving it a very non-academic and inappropriately informal casual feel in places.
7. There are also a few more minor concerns:
a. (2:65) “As further supported by findings…” What are the findings? The reference in this sentence to the previous section is not entirely clear.
b. (2:79) “The past decade has registered…” the sentence then refers to 2008, which is now 15 years ago.
c. (3:117) “It is safe to point towards residential satisfaction…” Is this really self-evident? Is there some data suggesting this is true? The City by Park and Burgess would suggest a much more complicated urban ecology and almost a century of work has built on that foundation.
d. (3:140-1) “…the city has been growing for the fact that the country is now able to grow economically…” I guess the sense here is that Angola has been able to grow economically since the end of the civil war, allowing the city to grow?
e. The second Table 1 (6:240-242) would benefit from comparative data from Alvalade.
f. (7:261) “…conducted with over the span of one week,” Conducted with whom?
g. (7:278-285) This seems highly sociological. This could be strengthened by connecting it more directly to a sociological literature.
h. (9:355) “Coup-de-sac” cul-de-sac?
i. Tables 2and 3 seem complicated and opaque. Is there some way to simplify them?
j. Earlier explanation of the term musseque and how it relates to and differs from favelo and slum would be helpful. It is ultimately answered, but the reader unfamiliar with Angolan urbanization is left too long in suspense.
k. Future reference to the work of Dan Silver (Scenescapes) may be useful to this sort of work and more useful references may be found in the journals Cities, Urban Affairs, Urban Studies, etc.
Author Response
Dear Reviewer, thank you very much for the reviews and advices. I carefully looked the suggested points and did my best to address them while maintaining consistency with the Theory and Data.
Please find here my responses to each one of the suggested points, as the Manuscript gets resubmitted as well.
----------------------------------------
The work seems interesting, entailed a lot of hard work, has some good data and I would ultimately like to see this work in publication. However, it has some clear issues that need to be addressed first, and I am not entirely convinced that this is the right venue for publication.
- What is the research question? There are points that suggest things like purpose, significance, and applications, but I was never clear with a clear and exact statement of the question the research is trying to answer. (23:812) mentions a hypothesis, but this is the first point at which it is mentioned in these clear terms.
There are two further related points that could help with the resolution of this issue:
- The need to connect the musseque to the neighborhood is not made clear.
- What is the connection to sustainability? Mention is made of SDG11, but the connection to the research is never made explicit as to how this research offers guidance in this direction. I can imagine connections but they need to be made clear and explicit, with implicit links seen throughout. I would suggest discussing SDG 11 and its component parts at the start, and them connect the parts of the study, organically, explicitly, and implicitly to the goal and its sub-goals.
Response 1: I addressed the issue and edited the text in order to mention purpose and hypothesis much earlier, that is, between 2:179 and 2:187, to expose them to the reader much earlier on the article, while establishing a connection between the Musseque and the city through the behavior at the border. On top of that, I further elaborated into the SDG 11 to provide more clarification.
- Why was Catambor selected as the location? 5:204 says, “And after deep analysis…” What analysis? What was the basis for the selection? Perhaps you could make more explicit the connection in 6:231-235 in the earlier discussion?
Response 2: I elaborated a bit more on the issue and I pointed some of the characteristics that lead to Catambor being chosen as the research Area. 5: 231 to 5:223.
- The numerous codes for the various behaviors become confusing. Either provide a table for the codes or use plain language descriptors in the text (with or without the accompanying code).
Response 3: I added further explanation about the codes in context, the first time they are directly mentioned, as well as when it isn’t clear from context which behavior is it.
- Figures 21, 22, and 24 all had “None Perceived” as by far the most popular choice. There needs to be more accounting for this in the text. The figures could be omitted, with mention of the apparent anomaly. Alternately, the results could be explained with a theoretical basis as to why the result is unobjectionable, or as to how and why the result is meaningful.
Response 4: I further developed the analysis to account with the lack of awareness and the absense of a clear opinion both on posiitve and negative way, and I believe it expanded on the relevance of the work indeed.
- The Discussion lacks a clear summary of the results. I hesitate to suggest more tables or figures, but that would be one way to streamline the presentation. Alternately, there could be a first paragraph summarizing the results. This might seem redundant following the summary of the interview results, but a single place that synthesizes the interviews and observations would clarify the results substantially.
Response 5: I edited the Discussion of the study in order to accommodate for a simplified version of the results as well, to better streamline the results.
- There are frequent English errors, some requiring a close reading for the correct understanding of the passage, in addition to numerous errors and typos (e.g. 4:182 “Rad” for “red”; 3:119 “pursue” for “pursuit”). There is a systematic issue with miscapitalization, with as well as frequent use of conjunctions at the beginning of sentences, giving it a very non-academic and inappropriately informal casual feel in places.
Response 6: I reviewed the paper once again and I did find numerous English errors and impropper uses of grammar, I tried to rewrite it more carefully and I asked did ask for a native speaker to review, with the remaining time I had. However, I would make use of an edition service in case its necessary.
- There are also a few more minor concerns:
- (2:65) “As further supported by findings…” What are the findings? The reference in this sentence to the previous section is not entirely clear.
- (2:79) “The past decade has registered…” the sentence then refers to 2008, which is now 15 years ago.
Response 7a,b: I further clarified the paragraph from 2:67 to 2:73. I reedited some of the refferences and updated some of the terminology that at the moment is simply outdated.
- (3:117) “It is safe to point towards residential satisfaction…” Is this really self-evident? Is there some data suggesting this is true? The City by Park and Burgess would suggest a much more complicated urban ecology and almost a century of work has built on that foundation.
Response 7c: I also reviewed the work of Park and Burgess and improved on the foundation behind the natural occupation and natural development of the city. It was indeed a very welcomed idea, as the theory of urban ecology gave a much more solid base for my theories for the natural development of setlements.
- (3:140-1) “…the city has been growing for the fact that the country is now able to grow economically…” I guess the sense here is that Angola has been able to grow economically since the end of the civil war, allowing the city to grow?
Response 7d: I managed to rephrase and re-reference the paragraph now between 4:165 and 4:169, which I believe gave it much more clarity.
- The second Table 1 (6:240-242) would benefit from comparative data from Alvalade.
Response 7e: I hesitated in providing data for Alvalade neighborhood as well, because despite the data being focused at the border, the study is centered on the fact tha the Catambor residents are the ones who cheate and shape the behavior on the street. So, for now I would like to keep the information a bit more streamlined but please note that I can ad it if necessary.
- (7:261) “…conducted with over the span of one week,” Conducted with whom?
Response 7f: I Rephrased many of the words we found on the
- (7:278-285) This seems highly sociological. This could be strengthened by connecting it more directly to a sociological literature.
Response 7g: I made use of more rereference regarding ethnography and sociological research to provide a proper base to the statement.
- (9:355) “Coup-de-sac” cul-de-sac?
- Tables 2and 3 seem complicated and opaque. Is there some way to simplify them?
- Earlier explanation of the term musseque and how it relates to and differs from favelo and slum would be helpful. It is ultimately answered, but the reader unfamiliar with Angolan urbanization is left too long in suspense.
- Future reference to the work of Dan Silver (Scenescapes) may be useful to this sort of work and more useful references may be found in the journals Cities, Urban Affairs, Urban Studies, etc.
Response 7h,I,j,k: I procedded with several corrections, but I also adressed the connections to the Favela, from 3:105 to 3:110, and I did read through the work of Dan Silver with scenescapes.
Reviewer 2 Report
Dear Authors,
I have reviewed the paper and found it interesting. However, this manuscript needs a lot of revision.
Comments:
1. Moderate English changes required.
2. The abstract of the article was written casually. The logic of the statement is not clear. Please rewrite it.
3. There are many errors in the format of the references in the article.
4. In the introduction the review of previous studies need to add in. To let readers know what research gaps this study fills.
5. Figure 2 should have some text in the figure to inform the reader of the location of the specific research site. It is difficult to express clearly simply by drawing.
6. The section of 2.1 Data Collection. If it is narrated separately and the collection purpose, collection method and analysis method of each data are explained clearly, it may be more helpful for readers to understand the research content.
7. The results and analysis section is a little confusing and not easy to read.
8. Section 3.6.2 – Reasons behind behavior and Impressions regarding the Environment.
The current research results are mainly descriptive, and some analysis should be added to make the research results statistically significant.
9. The discussion part should add some comparison with previous research results to discuss the new findings of this study.
Author Response
Dear Reviewer, thank you very much for the reviews and advices. I carefully looked the suggested points and did my best to address them while maintaining consistency with the Theory and Data.
Please find here my responses to each one of the suggested points, as the Manuscript gets resubmitted as well.
----------------------------------------
I have reviewed the paper and found it interesting. However, this manuscript needs a lot of revision.
Comments:
- Moderate English changes required.
Response 1: I reviewed the paper once again and I did find numerous English errors and impropper uses of grammar, I tried to rewrite it more carefully and I asked did ask for a native speaker to review, with the remaining time I had. However, I would make use of an edition service in case its necessary.
- The abstract of the article was written casually. The logic of the statement is not clear. Please rewrite it.
Response 2: I addressed the issue and rewrote the abstract, on a more concise manner.
- There are many errors in the format of the references in the article.
Response 3: I revied the refferences as well, and indeed there were numerous formating errors, I reformated it completely following the MDPI guidelines.
- In the introduction the review of previous studies need to add in. To let readers know what research gaps this study fills.
Response 4: I reviewed the introduction and added refferences to previous and comparative studies, including references to the Brazilian Favela and shantytowns in general, but also of street environment itself. In addition to that,
- Figure 2 should have some text in the figure to inform the reader of the location of the specific research site. It is difficult to express clearly simply by drawing.
Response 5: Thank you very much for the comment once again. I addressed the issue as well and added more description to the Figure.
- The section of 2.1 Data Collection. If it is narrated separately and the collection purpose, collection method and analysis method of each data are explained clearly, it may be more helpful for readers to understand the research content.
Response 6: I tried to add more clarity to the statements and completely rewrite what was unclear and inconcise.
- The results and analysis section is a little confusing and not easy to read.
Response 7: I further developed the section and added more interpretation of the results which was lacking before.
- Section 3.6.2 – Reasons behind behavior and Impressions regarding the Environment.
The current research results are mainly descriptive, and some analysis should be added to make the research results statistically significant.
Response 8: Similar to the 7th point, I tried to further develop the section and add some of my interpretation as well.
- The discussion part should add some comparison with previous research results to discuss the new findings of this study.
Response 9: I edited the Discussion of the study in order to accommodate for a simplified version of the results as well, to better streamline the results.
Round 2
Reviewer 1 Report
This version is substantially improved, though I have remaining concerns.
I still feel the research question could be made more explicit, but I will concede the point.
- The numerous codes for the various behaviors become confusing. Either provide a table for the codes or use plain language descriptors in the text (with or without the accompanying code).
Response 3: I added further explanation about the codes in context, the first time they are directly mentioned, as well as when it isn’t clear from context which behavior is it.
I still feel the tables are unclear with the codes, as well as in the text. The codes are probably more intuitive for a Portuguese speaker, but I think the tables and the text could replace the codes with plain language (with a code in parentheses or in a footnote where clarification is needed). My suggestions are: OR = hanging out; CS = consumption; TIS = socializing; CM = consumption; JD = games or playing games; and OT = other. These descriptors would be short and clear enough to fit in the tables as well as in the text and would significantly increase clarity.
- Figures 21, 22, and 24 all had “None Perceived” as by far the most popular choice. There needs to be more accounting for this in the text. The figures could be omitted, with mention of the apparent anomaly. Alternately, the results could be explained with a theoretical basis as to why the result is unobjectionable, or as to how and why the result is meaningful.
Response 4: I further developed the analysis to account with the lack of awareness and the absense of a clear opinion both on posiitve and negative way, and I believe it expanded on the relevance of the work indeed.
Appreciated. Further, would it be able to replace “None Perceived” with “No Opinion” as that would have a more neutral sound and would more accurately capture the sense of ethnographic conversations in the field.
- There are frequent English errors, some requiring a close reading for the correct understanding of the passage, in addition to numerous errors and typos (e.g. 4:182 “Rad” for “red”; 3:119 “pursue” for “pursuit”). There is a systematic issue with miscapitalization, with as well as frequent use of conjunctions at the beginning of sentences, giving it a very non-academic and inappropriately informal casual feel in places.
Response 6: I reviewed the paper once again and I did find numerous English errors and impropper uses of grammar, I tried to rewrite it more carefully and I asked did ask for a native speaker to review, with the remaining time I had. However, I would make use of an edition service in case its necessary.
The issue of miscapitalization is not addressed. I do not want to do all the work of going through the entire paper but have included the instances I noted from the first four pages. Italicized words should not be capitalized:
Looking at the relationship between General Urban growth, Economic development, 76
The decade of 2000 to 2010 has registered a high influx of people into Urban Areas, 80
having the Urban Population cross the 50% mark by 2008. Such growth is projected to 81
raise the Urban population up to 4.9 billion by 2030, while shrinking the Rural Population 82
by 28 million. This growth is being proven hard to deal with for Urban planners, as it 83
The rapid growth of cities had been linked to the birth of Slums in many different 87
The connection between rapid Urban growth and Slum development is par- 89
Urbanized region of the world. 91
90% of the increase of the World Urban Population by 2050 (15), suggests that the potential 93
Despite the rapid growth rate in Urban Population, the proportion of Slum dwellers 95
relative to the Urban population in general, is in fact, in decline, but that decline does not 96
to 61.7%, signaling a reduction relative to its total Urban population, whilst the number 99
tion growth within slums but an even higher Urban population growth in general. infer- 101
ences in the Architectural space and Urban landscape, that ultimately translate to differ- 122
to classify and shed a light in the Environmental Behavior itself, for they teach us what 125
problems in the urban context as a part of Urban regeneration itself (21). However, imple- 152
redesigning of complete areas of a city is often considered to the primary go-to by Urban 154
Planners, can pose unique challenges and raise questions about which elements of the 155
With the current Urban development rates in Sub-Saharan Africa, those settlements 184
seem to be directly associated with the current rates of Urban Growth (23). And given the 185
Unplanned nature of the settlements, there is a vast number of different characterizations 186
seems to do the best job encompassing the differences between formal Urbanization and 190
the informal development directly associated with Slums (24). 191
In addition, there continue to be frequent use of casual conjunctions (And, But) at the beginning of sentences as is common in informal communications but is only used to establish an informal tone or make a particular point in formal writing. “And” can be replaced with “In addition,” “Moreover,” or “Furthermore,” while “But” can be replaced by “However,” “In contrast,” or “On the other hand,” in order to give the paper a more formal and authoritative tone.
- The second Table 1 (6:240-242) would benefit from comparative data from Alvalade.
Response 7e: I hesitated in providing data for Alvalade neighborhood as well, because despite the data being focused at the border, the study is centered on the fact tha the Catambor residents are the ones who cheate and shape the behavior on the street. So, for now I would like to keep the information a bit more streamlined but please note that I can ad it if necessary.
I understand your concerns. Is there some way to convey the contrast between the neighborhoods so that the boundary is salient? I have never been to Luanda so I don’t know when Alvalade was created, how populous it is, how many people there are in a household, how much electricity they have, how much access to water, housing construction. As far as I know, Catambor, with 75% electricity, may be the same as Alvalade. The boundary is different if the two neighborhoods are more similar than if they are more different. A comparison in the table seemed the easiest way to convey this, but I would be satisfied if there is some other way.

Author Response
Response to Reviewer 1 Comments
Dear Reviewer, thank you very much for the reviews and advices. I carefully looked the suggested points and did my best to address them while maintaining consistency with the Theory and Data.
Please find here my responses to each one of the suggested points, as the Manuscript gets resubmitted as well.
----------------------------------------
- I still feel the tables are unclear with the codes, as well as in the text. The codes are probably more intuitive for a Portuguese speaker, but I think the tables and the text could replace the codes with plain language (with a code in parentheses or in a footnote where clarification is needed). My suggestions are: OR = hanging out; CS = consumption; TIS = socializing; CM = consumption; JD = games or playing games; and OT = other. These descriptors would be short and clear enough to fit in the tables as well as in the text and would significantly increase clarity.
Response 1: I changed some of the language around the tables and made sure that the language was clear enough.
- Appreciated. Further, would it be able to replace “None Perceived” with “No Opinion” as that would have a more neutral sound and would more accurately capture the sense of ethnographic conversations in the field.
Response 2: Thank you very much for the comments, I addressed that as well, in both graphics and text.
- The issue of miscapitalization is not addressed. I do not want to do all the work of going through the entire paper but have included the instances I noted from the first four pages. Italicized words should not be capitalized: (…)
Response 3: I also reviewed the entire document, thank you very much for the comments.
3.1 In addition, there continue to be frequent use of casual conjunctions (And, But) at the beginning of sentences as is common in informal communications but is only used to establish an informal tone or make a particular point in formal writing. “And” can be replaced with “In addition,” “Moreover,” or “Furthermore,” while “But” can be replaced by “However,” “In contrast,” or “On the other hand,” in order to give the paper a more formal and authoritative tone.
Response 3.1: I addressed the issue with the casual conjunctions while reviewing the whole doccument as well.
- I understand your concerns. Is there some way to convey the contrast between the neighborhoods so that the boundary is salient? I have never been to Luanda so I don’t know when Alvalade was created, how populous it is, how many people there are in a household, how much electricity they have, how much access to water, housing construction. As far as I know, Catambor, with 75% electricity, may be the same as Alvalade. The boundary is different if the two neighborhoods are more similar than if they are more different. A comparison in the table seemed the easiest way to convey this, but I would be satisfied if there is some other way.
Response 4: About the Alvalade neighborhood, I managed to add more information about location, comodities and origins, for contextualization, please make sure to comment on whether or not it is adequate, in the end.
Again, thank you very much for the comments.

Reviewer 2 Report
The article has undergone a lot of effective revisions, and I think it can be published after minor revisions.
1. The abstract needs to be further modified, and important results should be written in the abstract.
2. The result part has been written clearly, but there are many redundant sentences. It is suggested to simplify it.
3. The discussion part needs to be further improved. This part should explain the similarities and differences between this study and previous studies, and highlight the significance of the study.
Author Response
Dear Reviewer, thank you very much for the reviews and advices. I carefully looked the suggested points and did my best to address them while maintaining consistency with the Theory and Data.
Please find here my responses to each one of the suggested points, as the Manuscript gets resubmitted as well.
----------------------------------------
The article has undergone a lot of effective revisions, and I think it can be published after minor revisions.
Comments:
- The abstract needs to be further modified, and important results should be written in the abstract.
Response 1: I managed to review the abstract once again and added further mentions of important results in a simplified manner.
- The result part has been written clearly, but there are many redundant sentences. It is suggested to simplify it.
Response 2: I attemped to reduce some of the redundancies within the results part. It was still a bit difficult to chose. Between things to remove and to keep but I tried my best to make it simple but still maintain the relevancy of the analyzed work.
- The discussion part needs to be further improved. This part should explain the similarities and differences between this study and previous studies, and highlight the significance of the study.
Response 3: I added further explanation on the problem and significance of the study, and tried to ad as well forther reviews on works that preceeded and are relevant for the study.
